# DIFFERENTIATION THROUGH BLACK-BOX QUADRATIC PROGRAMMING SOLVERS

## ABSTRACT

In recent years, many deep learning approaches have incorporated layers that solve optimization problems (*e.g.*, linear, quadratic, and semidefinite programs). Integrating these optimization problems as differentiable layers requires computing the derivatives of the optimization problem's solution with respect to its objective and constraints. This has so far limited the use of state-of-the-art black-box numerical solvers within neural networks, as they lack a differentiable interface. To address this issue for one of the most common convex optimization problems – quadratic programming (QP) – we introduce **dQP**, a modular framework that enables plug-and-play differentiation for any QP solver, allowing seamless integration into neural networks and bi-level optimization tasks. Our solution is based on the core theoretical insight that knowledge of the active constraint set at the QP optimum allows for *explicit* differentiation. This insight reveals a unique relationship between the computation of the solution and its derivative, enabling efficient differentiation of any solver, that only requires the primal solution. Our implementation, which will be made publicly available upon acceptance, interfaces with an existing framework that supports over 15 state-of-the-art QP solvers, providing each with a fully differentiable backbone for immediate use as a differentiable layer in learning setups. To demonstrate the scalability and effectiveness of dQP, we evaluate it on a large benchmark dataset of QPs with varying structures. We compare dQP with existing differentiable QP methods, demonstrating its advantages across a range of problems, from challenging small and dense problems to large-scale sparse ones, including a novel bi-level geometry optimization problem.

## 1 INTRODUCTION

Computational methods rely heavily on solving *optimization problems*, *i.e.*, finding an optimum of a given function, under some given constraints on the solution. Optimization is arguably the most popular method to approach computational problems that do not admit a closed-form solution. This in turn has led to the development of both open-source and commercial numerical solvers specialized for different classes of optimization, especially constrained convex optimization (Wright, 2006; Boyd & Vandenberghe, 2004). It is, thus, quite enticing to incorporate optimization as a "layer" within machine learning architectures, *e.g.*, where a neural network's intermediate output defines the optimization problem, and the solution of that optimization problem is taken as the final output of the neural network (Amos & Kolter, 2017; Agrawal et al., 2019a; Blondel & Roulet, 2024). This approach has proven successful on many practical tasks including image classification (Amos et al., 2017), optimal transport (Rezende & Racanière, 2021; Richter-Powell et al., 2021), zero-sum games (Ling et al., 2018), tessellation (Chen et al., 2022), control (Amos et al., 2018; de Avila Belbute-Peres et al., 2018; Ding et al., 2024), decision-making (Tan et al., 2020), robotics (Holmes et al., 2024), biology (Zhang et al., 2023), and natural language processing (Thayaparan et al., 2022).

In general, training a neural network requires the ability to *backpropagate* gradients to optimize the network's weights and biases. Hence, in case the network includes an optimization layer as described above, one needs to have a way to *differentiate* that layer, *i.e.*, compute the gradients of the solution of the optimization problem with respect to the parameters of the optimization problem itself. Gradients can be obtained through optimality conditions, which provide a characterization that allows for the application of the implicit function theorem (Krantz & Parks, 2012) and implicit differentiation. However, this approach requires the dual solution and yields a linear system

for the gradients that can be costly to invert for large problems. As a result, previous differentiable methods tightly couple the differentiation to a custom tailor-made solver that outputs a dual solution, allows information from the solution algorithm to be re-used, or uses GPU acceleration.

The existing tight coupling between neural architecture and optimizer severely limits the applicability of neural optimization methods: in general, solving optimization problems is a hard, challenging task, requiring state-of-the-art solvers such as Gurobi (Gurobi Optimization, LLC, 2024) and MOSEK (Andersen & Andersen, 2000) which have been developed through years of commercial and academic research. These solvers provide the capability to efficiently and reliably handle problems at a scale that non-optimized implementations cannot achieve. More so, even having one of these solvers at hand is not enough, since none of them is a "catch-all" solution. Instead, it is necessary to choose and swap between specific solvers for specific structures of optimization problems that may emerge in different learning tasks. To solve this issue and obtain a general, efficient way to interface between general solvers and neural networks via backpropgation, one needs to devise a "bridge" to differentiate through these blackbox solvers.

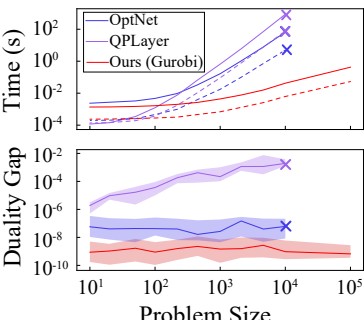

Figure 1: Comparison over QP problems of increasing size. Ours (using Gurobi) outperforms OptNet and QPLayer in terms of forward time (solid), backward time (dashed), and accuracy as problem size increases. OptNet and QPLayer become intractable for larger problems.

In this paper, we focus on completely removing the above limitation to one of the most canonical and important convex optimization problems, quadratic programing (QP), which minimize a quadratic objective under linear inequality constraints. Our framework, which we dub **dQP** (as in differential notation), augments any QP solver into a differentiable one and seamlessly integrates it as a differentiable layer. dQP stems from our main novel theoretical observation: the gradients of the optimal point of a QP can be obtained *explicitly* from a primal solution provided by the optimizer, and the active set of constraints, which can be deduced from the solution.

We draw this conclusion by leveraging classic sensitivity theory, and by clarifying the role of the active set which has otherwise appeared in recent work, but without emphasis and not in this form. Notably, our explicit perspective recasts the traditional implicit differentiation approach into an *explicit* method, which provides a straightforward pathway to complete solver modularity. Namely, we avoid the costly linear solve for the necessary gradients by showing the full gradients can be expressed solely through the (much smaller) active set. Furthermore, we show that this reduced system can also be used to solve for the active dual variables, if not provided by a solver. This enables us to implement dQP on top of a minimal open-source interface (Caron et al., 2024b), which provides direct access to over 15 free and commercial QP solvers and easily supports the integration of additional solvers.

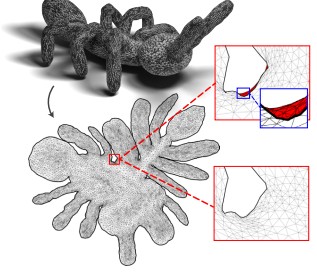

Figure 2: dQP solves a bi-level optimization problem to compute the bijective planar embedding of a large-scale ant mesh (15k vertices).

Using the modularity of our method, we conducted a comprehensive evaluation on a diverse benchmark dataset comprising over 2,000 QPs, comparing dQP's performance against existing differentiable QP methods. As highlighted in Figure 1, dQP demonstrates a significant advantage in structured QPs when paired with state-of-the-art sparse QP solvers. We show the superiority of our method on tasks such as sparse projections, as well as in a novel geometric bi-level optimization experiment that was intractable for previous methods, whereas our method excels.

To summarize, our contributions are:

1. We prove that QPs can be explicitly differentiated using only the primal solution via a locally-equivalent linear system.

2. Building on this, we devise and implement a fully modular differentiable layer compatible with any QP solver, allowing for plug-and-play flexibility where users can easily select the best solver for their specific task. Our open-source implementation will be made publicly available.

3. We demonstrate state-of-the-art performance in solving and differentiating large-scale, sparse QPs using various solvers across a series of extensive experiments.

## 2 RELATED WORKS

**Implicit Layers.** Optimization layers are an example of recently introduced implicit layers that leverage implicit differentiation to compute gradients of solution mappings without requiring closed-form expressions (Duvenaud et al., 2020). This category also includes deep equilibrium models, which represent fixed-point mappings and can be viewed as networks of infinite depth (Bai et al., 2019; Kawaguchi, 2021; El Ghaoui et al., 2021; Gurumurthy et al., 2021; Winston & Kolter, 2020; Bai et al., 2020). Extending beyond algebraic equations, similar techniques are applied in neural ordinary differential equations using the adjoint state method from parametric control for partial differential equations (Lions, 1971; Xue et al., 2020; Beatson et al., 2020; Chen et al., 2018). Implicit differentiation is also used in bi-level programming where optimization problems are nested in one-another (Colson et al., 2007; Kunisch & Pock, 2013; Gould et al., 2016; Alesiani, 2023) and meta-learning where the outer learning process is optimized (Finn, 2018; Andrychowicz et al., 2016; Hochreiter et al., 2001; Hospedales et al., 2021; Rajeswaran et al., 2019; Sambharya et al., 2024). Alternative approaches avoid implicit differentiation by using approximation techniques. For instance, they apply automatic differentiation directly to iterative algorithms through loop unrolling (Belanger & McCallum, 2016; Belanger et al., 2017; Metz et al., 2019; Scieur et al., 2022) or, in the case of fixed-point mappings, by differentiating a single iteration or employing a Jacobian-free method (Geng et al., 2021; Fung et al., 2022; Bolte et al., 2023).

**Sensitivity Analysis and Parametric Programming.** There is extensive mathematical theory on the local behavior of optimization problems under perturbations (Rockafellar, 1970; Rockafellar & Wets, 1998), particularly in assessing the sensitivity and stability of solutions (Fiacco, 1983; 1990; Fiacco & McCormick, 1968; Lee et al., 2010; Bonnans & Shapiro, 2013). For this, the implicit function theorem is a central analytical tool, but unlike fixed point mappings, an intermediate step is required before it can be applied. Particularly, one must first pass from the optimization problem itself to its optimality conditions, often requiring a number of assumptions in order for them to be completely equivalent. The theoretical results of sensitivity theory underpin applications in multi-parametric programming (Pistikopoulos et al., 2020), like model predictive control, where the problem is solved for various input parameters, leading to intense computations. To address this, Bemporad et al. (2002) observed that QP systems have a closed-form solution if the active or binding set is known beforehand, allowing it to be pre-computed offline. This approach requires partitioning the parameter space into regions of fixed active set (Spjøtvold et al., 2006), inside which the active set is stable to perturbations. Methods based on this idea continue to be developed for solving parametric QPs (Ferreau et al., 2014; Narciso et al., 2022; Arnström & Axehill, 2024).

**Differentiable Programming.** Our work follows OptNet (Amos & Kolter, 2017; Amos, 2019) which differentiates QPs through their optimality conditions, and focuses on small dense problems for GPU batching. They solve the full Jacobian system efficiently by reusing the factorization employed in their custom interior-point method. However, as noted in (Bambade et al., 2024), this comes at the cost of ill-conditioning due to symmetrization. More recent differentiable QP methods include Alt-Diff and SCQPTH (Sun et al., 2022; Butler & Kwon, 2023; Butler, 2023) which use first-order ADMM and approximately differentiate the fixed point map, and QPLayer (Bambade et al., 2024) focusing on accommodating infeasibility via extended conservative Jacobians. Similarly to OptNet, several other solvers are tightly integrated with specific algorithms, often to enable access to internal computations required for differentiation. Alt-Diff is coupled with a custom ADMM method, SCQPTH reimplements OSQP, and QPLayer is built on ProxQP. Several works have highlighted the importance of the active constraint set in differentiating constrained optimization problems (Amos et al., 2017; Gould et al., 2022; Paulus et al., 2021), as well as in the context of quadratic programming (Amos et al., 2018; Bambade et al., 2024; Pan et al., 2024; Niculae et al., 2018). A common observation is that the algebraic system obtained through implicit differentiation can be simplified by removing rows corresponding to inactive constraints. Amos et al. (2017); Pan et al. (2024) have additionally observed that backpropagation can be cast as an equality-constrained QP parameterized by incoming gradients. However, existing approaches do not directly utilize the formation of a significantly dimension-reduced symmetric linear system to efficiently differentiate arbitrary black-box QP solvers, thus missing the opportunity to effectively decouple

optimization and differentiation. Other classes of optimization problems, such as convex cone programs (Agrawal et al., 2019b) and mixed-integer programs (Paulus et al., 2021), have also been differentiated. Frameworks proposed in (Agrawal et al., 2019a; Blondel et al., 2022; Pineda et al., 2022; Besançon et al., 2024; Paulus et al., 2024) provide a differentiable interface to broader classes of optimization problems, with QP as a subset. However, CVXPYLayers (Agrawal et al., 2019a) reformulates the QP into a cone program to utilize diffcp internally (Agrawal et al., 2019b). As a result, it does not support specialized QP solvers and instead relies exclusively on the cone solvers SCS, ECOS, and Clarabel. The framework Theseus (Pineda et al., 2022) directly handles only unconstrained problems and similarly lacks support for QP-specific solvers. JAXopt (Blondel et al., 2022) includes a differentiable reimplementation of OSQP and an implicit differentiation wrapper for CVXPY, which requires symbolic compilation of the QP. Both CVXPYLayers with diffcp and JAXopt with a QP solver necessitate the entire primal-dual solution to construct the linear system for derivatives obtained via implicit differentiation. Additionally, both frameworks demonstrated subpar performance compared to the specialized QPLayer, as reported in (Bambade et al., 2024).

## 3 APPROACH

We now detail the theoretical underpinning of our method that can transform any black-box QP solver into a differentiable layer. We begin by formulating the problem concretely, move on to establishing basic theory of differentiation of QP's (via sensitivity analysis and KKT conditions), and finally connect those with our main novel theoretical observation, leading to a straightforward algorithm. We note that various subparts of the theoretical background discussed in the following have been used in recent years to develop differentiable QP layers, see Section 2. However, no single work has fully leveraged this theory to completely decouple optimization and differentiation in a manner that supports arbitrary state-of-the-art QP solvers while also ensuring efficient, robust differentiation.

### 3.1 PROBLEM SETUP: DIFFERENTIATING QUADRATIC PROGRAMS

We consider a quadratic program which is feasible and strictly convex (*i.e.*, $P \succ 0$) in standard form,

$$
\begin{aligned}
z^*(\theta) = \arg\min_z \quad & \frac{1}{2} z^T P(\theta) z + q(\theta)^T z \\
\text{subject to} \quad & A(\theta) z = b(\theta) \\
& C(\theta) z \leq d(\theta),
\end{aligned}
\tag{1}
$$

where $P \in \mathbb{R}^{n \times n}, q \in \mathbb{R}^n, A \in \mathbb{R}^{p \times n}, b \in \mathbb{R}^p, C \in \mathbb{R}^{m \times n}$ and $d \in \mathbb{R}^m$ are smoothly parameterized by some $\theta \in \mathbb{R}^s$. This $\theta$ can either be the output of a previous layer in a neural network, or otherwise learnable parameters. To simplify notation, in the following we omit $\theta$.

To motivate our work, consider the case in which a QP of the form Equation (1) is incorporated as the $\ell$-th layer of a neural network, *i.e.*, the QP layer receives an input vector $x_\ell$ and outputs a vector $x_{\ell+1}$ satisfying the relation $x_{\ell+1} = z^*(x_\ell)$. In other words, the QP layer's input, $x_\ell$, serves as the parameters $\theta$ that control the objective and constraints of the QP, and the optimal point $z^*(x_\ell)$ is the layer's output. Training a neural network requires *backpropagating* gradients through it, which involves computing the Jacobian of the layer's output with respect to its input, $\frac{\partial x_{\ell+1}}{\partial x_\ell}$. In the case of a QP layer, these gradients are exactly $\frac{\partial z^*}{\partial \theta}$, *i.e.*, the derivative of the optimal point $z^*$ with respect to the parameters $\theta$. The same derivative is also essential when using descent methods to solve certain bi-level optimization problems (Colson et al., 2007).

In this work, we focus on computing $\partial_\theta z^*(\theta) = \frac{\partial}{\partial \theta} z^*(\theta)$, the derivative of the optimal point of the QP Equation (1) with respect to the parameters $\theta$. Intuitively, this derivative quantifies the change in the optimal point of the QP in response to a perturbation of its parameters $\theta$. Our goal is to efficiently compute $\partial_\theta z^*(\theta)$ independently of the method used to approximate the optimal point $z^*(\theta)$.

### 3.2 THEORETICAL DIFFERENTIATION OF QPS VIA KKT CONDITIONS AND SENSITIVITY

Our goal is to devise a method for differentiating QPs based solely on the solution provided by a black-box numerical solver. First, we need to establish the *theoretical* foundations necessary for the

desired derivatives. These derivatives, as is common in optimization, are obtained through *sensitivity analysis* applied to the *KKT conditions*. In this section, we elaborate on these concepts, synthesizing key theoretical insights from optimization, sensitivity analysis, parametric programming, and differentiable programming techniques, distilling them in the context of QPs to lay the groundwork for our results and the development of dQP.

**Optimality Conditions.** The first-order Karush–Kuhn–Tucker (KKT) conditions (Karush, 1939; Kuhn & Tucker, 1951; Boyd & Vandenberghe, 2004; Wright, 2006) provide a useful algebraic characterization of the optimal points of constrained optimization problems. In essence, they are an extension of the method of Lagrange multipliers for problems that include inequalities. For the QP Equation (1), the KKT conditions take the form,

$$
\begin{aligned}
Pz^* + q + A^T\lambda^* + C^T\mu^* &= 0 \\
Az^* - b &= 0 \\
Cz^* - d &\leq 0 \\
\mu^* &\geq 0 \\
D(\mu^*)(Cz^* - d) &= 0,
\end{aligned}
\tag{2}
$$

where $D(\mu^*) = \mathrm{diag}(\mu^*)$, and the additional added variables $\lambda^* \in \mathbb{R}^p$ and $\mu^* \in \mathbb{R}^m$ are called *the optimal dual variables* of the linear equalities and inequalities, respectively. With these dual variables, one considers the extended *primal-dual solution* $\zeta^*(\theta) = (z^*(\theta), \lambda^*(\theta), \mu^*(\theta))$ of the QP Equation (1). Crucially, under strict convexity and feasibility, the QP Equation (1) has a unique solution $\zeta^*(\theta)$, and the KKT conditions Equation (2) are necessary and sufficient for its optimality.

**Active Set and Complementary Slackness.** A main point of interest in this work lies in the last equation of Equation (2), which is the nonlinear *complementary slackness* condition. Intuitively, it encodes the two situations in which each original inequality constraint from Equation (1), $(Cz^* - d)_j \leq 0$, may be. Either (1) the constraint is *active*, *i.e.*, it is satisfied as an equality $(Cz^* - d)_j = 0$, in which case $\mu_j^* \geq 0$; or (2) the constraint is *inactive*, *i.e.*, it is satisfied with a strict inequality, in which case $\mu_j^* = 0$. Importantly, an inactive constraint implies that the same optimal solution $z^*$ would be obtained even if that specific constraint were removed from the QP. We denote by $J(\theta) = \left\{ j : (C(\theta)z^*(\theta) - d(\theta))_j = 0 \right\}$ the set of active constraints.

**Derivatives via Sensitivity Analysis.** To define derivatives of QPs, we turn to the Basic Sensitivity Theorem (Theorem 2.1 in Fiacco (1976)), which provides the foundation for differentiating the KKT conditions with respect to $\theta$. To differentiate at $\theta$, the theorem requires the additional condition of *strict* complementary slackness; this prohibits the degenerate case where both $(Cz^* - d)_j = 0$ and $\mu_j^* = 0$, ensuring that a small perturbation of the parameters does not alter the active set. Under strict complementary slackness, it establishes that *in a neighborhood of $\theta$*, the primal-dual point $\zeta^*(\theta) = (z^*(\theta), \lambda^*(\theta), \mu^*(\theta))$ is a differentiable function of $\theta$, optimal for the QP Equation (1), uniquely satisfies the KKT conditions Equation (2), and maintains strict complementary slackness. Crucially, the active set $J(\theta)$ remains fixed within this neighborhood.

Since the active set is stable, the equality conditions in Equation (2) suffice to provide a local characterization of $\zeta^*(\theta)$. Implicit differentiation of these yields the Jacobians of the solution $\partial_\theta \zeta^*$ in terms of the following linear system,

$$
\begin{bmatrix} P & A^T & C^T \\ A & 0 & 0 \\ D(\mu^*)C & 0 & D(Cz^* - d) \end{bmatrix} \begin{bmatrix} \partial_\theta z^* \\ \partial_\theta \lambda^* \\ \partial_\theta \mu^* \end{bmatrix} = - \begin{bmatrix} \partial_\theta P z^* + \partial_\theta q + \partial_\theta A^T \lambda^* + \partial_\theta C^T \mu^* \\ \partial_\theta A z^* - \partial_\theta b \\ D(\mu^*)(\partial_\theta C z^* - \partial_\theta d) \end{bmatrix}.
\tag{3}
$$

Under the conditions for the Basic Sensitivity Theorem, the linear system Equation (3) is invertible. It degenerates exactly in the presence of weakly active constraints $\mu_j^* = (Cz^* - d)_j = 0$, for which the QP is non-differentiable (see, *e.g.*, Amos & Kolter (2017)). For any inactive constraint $j \notin J$, the dual variable $\mu_j^*$ vanishes, and thus the corresponding rows and columns of Equation (3) can be removed, simplifying it into the reduced form

$$
\begin{bmatrix} P & A^T & C_J^T \\ A & 0 & 0 \\ C_J & 0 & 0 \end{bmatrix} \begin{bmatrix} \partial_\theta z^* \\ \partial_\theta \lambda^* \\ \partial_\theta \mu_J^* \end{bmatrix} = - \begin{bmatrix} \partial_\theta P z^* + \partial_\theta q + \partial_\theta A^T \lambda^* + \partial_\theta C_J^T \mu_J^* \\ \partial_\theta A z^* - \partial_\theta b \\ \partial_\theta C_J z^* - \partial_\theta d_J \end{bmatrix},
\tag{4}
$$

where $\mu_J^*$, $C_J$ and $d_J$ denote restriction to rows corresponding to active inequality constraints $j \in J$.

## 3.3 EXTRACTING DERIVATIVES FROM A QP SOLVER'S SOLUTION

Through the above theory, we can obtain our main theoretical results and introduce **dQP**, a straightforward algorithm for efficient and robust differentiation of any black-box QP solver.

Our approach stems from two straightforward yet powerful insights: (1) given the *primal* solution of a QP, it is easy to identify the active set of a QP; (2) once the active set is known, both the primal-dual optimal point *and* its derivatives can be explicitly derived in closed-form. Furthermore, the computation of these quantities can then be achieved efficiently using a single matrix factorization of a reduced-dimension symmetric matrix.

These observations in turn lead to a simple algorithm that is easy to implement: first, solve the optimization problem using *any* QP solver; then, use the solution to identify the active set and solve a linear system to compute the derivatives. Consequently, we can define a "backward pass" for any layer that uses a QP solver, allowing for the seamless integration of any solver best suited to the problem, thus leveraging years of research and development invested in state-of-the-art QP solvers.

**Explicit Active Set Differentiation.** Consider a QP and its optimal point $\zeta^*(\theta)$, along with the set $J(\theta)$ of active constraints (see Section 3.2). We define the reduced equality-constrained quadratic program, obtained by removing inactive inequalities and converting active inequality constraints into equality constraints,

$$
\begin{aligned}
z^*(\theta) = \arg\min_z \quad & \frac{1}{2} z^T P(\theta) z + q(\theta)^T z \\
\text{subject to} \quad & \begin{bmatrix} A(\theta) \\ C(\theta)_{J(\theta)} \end{bmatrix} z = \begin{bmatrix} b(\theta) \\ d(\theta)_{J(\theta)} \end{bmatrix}.
\end{aligned} \tag{5}
$$

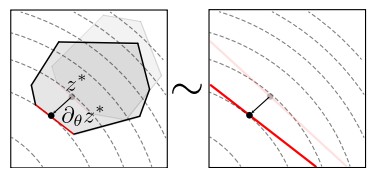

Figure 3: Schematic active set differentiation. Left: a QP is shown by its quadratic level sets and polyhedral feasible set; the solution lies on a facet of the boundary; perturbations of the constraints lead to perturbations in the solution. Right: the perturbation of the solution remains the same when inactive constraints are eliminated.

Under the assumptions of Section 3.2, this simpler QP is, in fact, *locally* equivalent to the QP Equation (1), as illustrated in Figure 3 with a 2D example. Moreover, it provides an explicit expression for both the primal-dual optimal point and its derivatives:

**Theorem 1.** *The QP Equation (5) is locally equivalent to the reduced equality-constrained QP Equation (1) and its solution* $\zeta^*(\theta) = (z^*(\theta), \lambda^*(\theta), \mu^*(\theta))$ *admits the explicit form*

$$
\begin{bmatrix} z^* \\ \lambda^* \\ \mu_J^* \end{bmatrix} = \begin{bmatrix} P & A^T & C_J^T \\ A & 0 & 0 \\ C_J & 0 & 0 \end{bmatrix}^{-1} \begin{bmatrix} -q \\ b \\ d_J \end{bmatrix}. \tag{6}
$$

*Furthermore, the optimal point can be explicitly differentiated to obtain*

$$
\begin{bmatrix} \partial_\theta z^* \\ \partial_\theta \lambda^* \\ \partial_\theta \mu_J^* \end{bmatrix} = - \begin{bmatrix} P & A^T & C_J^T \\ A & 0 & 0 \\ C_J & 0 & 0 \end{bmatrix}^{-1} \left( \begin{bmatrix} \partial_\theta P & \partial_\theta A^T & \partial_\theta C_J^T \\ \partial_\theta A & 0 & 0 \\ \partial_\theta C_J & 0 & 0 \end{bmatrix} \begin{bmatrix} z^* \\ \lambda^* \\ \mu_J^* \end{bmatrix} - \begin{bmatrix} -\partial_\theta q \\ \partial_\theta b \\ \partial_\theta d_J \end{bmatrix} \right). \tag{7}
$$

A proof of this Theorem, based on the Basic Sensitivity Theorem (Fiacco, 1976), is provided in Appendix A, along with a calculation of the derivatives using differential matrix calculus (Petersen & Pedersen, 2008; Magnus & Neudecker, 1988). We note that this result is closely related to analyses studied in multi-parametric programming (Bemporad et al., 2002; Pistikopoulos et al., 2020; Spjøtvold et al., 2006; Arnström & Axehill, 2024; Narciso et al., 2022).

Notably, in the case of quadratic programming, the Basic Sensitivity Theorem allows one to bypass the need for implicit differentiation techniques (Krantz & Parks, 2012). It is important to emphasize that this observation does not change the fact that the general solution and the corresponding active set lack a closed-form expression. Moreover, while we perform explicit differentiation, the implicit function theorem remains key in establishing the local equivalence between the two problems. The derivatives are indeed the same, and we do not suggest otherwise. However, the derivations to find them differ. The derivatives in Equation (7) are obtained by ordinary (explicit) differentiation of the

---

**Algorithm 1 – dQP**: Differentiation through Black-box Quadratic Programming Solvers

**Input:** $P, q, A, b, C, d$, and tolerance $\epsilon_J$
**Output:** $z^*, \lambda^*, \mu^*$ and $\partial_\theta z^*, \partial_\theta \lambda^*, \partial_\theta \mu^*$
  1: Solve QP Equation (1) with any solver for the primal solution $z^*$ (and $\lambda^*, \mu^*$ if available)
  2: Compute the active set by hard thresholding with tolerance: $J = \{j : (Cz^* - d)_j \geq -\epsilon_J\}$
  3: Factorize the reduced KKT system matrix: $K_J = \begin{bmatrix} P & A^T & C_J^T \\ A & 0 & 0 \\ C_J & 0 & 0 \end{bmatrix}$
  4: Compute $\lambda^*, \mu^*$ (if not obtained in step (1)): $\begin{bmatrix} z^* \\ \lambda^* \\ \mu_J^* \end{bmatrix} = K_J^{-1} \begin{bmatrix} -q \\ b \\ d_J \end{bmatrix}$
  5: Compute the derivatives: $\begin{bmatrix} \partial_\theta z^* \\ \partial_\theta \lambda^* \\ \partial_\theta \mu_J^* \end{bmatrix} = -K_J^{-1} \left( \begin{bmatrix} \partial_\theta P & \partial_\theta A^T & \partial_\theta C_J^T \\ \partial_\theta A & 0 & 0 \\ \partial_\theta C_J & 0 & 0 \end{bmatrix} \begin{bmatrix} z^* \\ \lambda^* \\ \mu_J^* \end{bmatrix} - \begin{bmatrix} -\partial_\theta q \\ \partial_\theta b \\ \partial_\theta d_J \end{bmatrix} \right)$

---

closed-form solution to the reduced QP Equation (6). On the other hand, the ones in Equation (4) are obtained by implicit differentiation of the original nonlinear KKT Equation (2) and followed by eliminating inactive rows. This perspective and Theorem 1 underscore a critical computational insight: once a black-box solver provides the primal solution to the QP, the active set can be determined, and additionally the derivatives can be computed via Equation (7). Furthermore, if the solver provides only the primal solution and not the primal-dual pair, the dual can be completed through Equation (6). Since the computation of the derivatives in Equation (7) requires the factorization of the KKT matrix $K_J$, completing the primal-dual solution through Equation (6) adds negligible computational cost – a single factorization produced by any direct solver (*e.g.*, from SuperLU (Li, 2005)) can be thus be used for both completing the dual solution via Equation (6) and computing the derivatives in Equation (7). All these insights lead up to the key algorithm of dQP, summarized in Algorithm 1.

**Numerical Computation.** Our approach leads to a compact and efficient computation of gradients. Indeed, the linear system Equation (7) that we factorize to compute the derivatives and dual solution is symmetric and reduced in size. In contrast, implicit differentiation of the full KKT conditions Equation (1) leads to a significantly larger, asymmetric system Equation (3). Beyond simplifying the derivative computation, our approach enables the use of fast, specialized linear solvers that exploit the reduced systems symmetric indefinite KKT matrix structure (*e.g.*, using an LDL factorization as in QDLDL (Stellato et al., 2020; Davis, 2005)).

Empirically, we observe that the reduced linear system Equation (7) is often significantly better conditioned than its full counterpart. The inset figure illustrates this with an example of a QP governed by two parameters $\theta = (\theta_1, \theta_2)$ from Spjøtvold et al. (2006), computed using DAQP (Arnström et al., 2022). The figure visualizes (a) regions in which the active set is constant, and (b) the conditioning of the full and reduced linear systems along a cross-section in parameter space (right), demonstrating that eliminating inactive constraints improves conditioning.

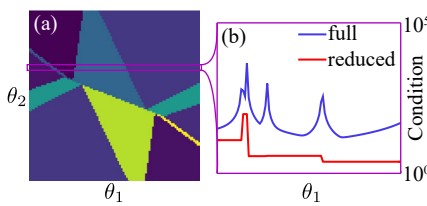

This figure also highlights the challenge of calculating derivatives near singularities, where the active set changes and some inequalities turn into weakly active, leading to ill-defined derivatives. Near such singularities, implicit differentiation suffers from severe ill-conditioning. This affects our approach as well, manifesting in the challenge of determining the active set at an approximate solution. Various methods have been proposed to address this issue, such as specialized algorithms for active set identification (Cartis & Yan, 2016; Oberlin & Wright, 2006; Burke & Moré, 1988). Our implementation includes an optional heuristic for active set refinement to address this instability, described in Appendix B. However, we found that simple hard thresholding of the primal residual $r_j = (Cz^* - d)_j \geq -\epsilon_J$ was sufficiently robust in all of our experiments, as shown in Section 4.

**Implementation.** Our open-source implementation will be made publicly available. We implement **dQP**, Algorithm 1, as a fully differentiable module in PyTorch (Paszke et al., 2019), providing a simple-to-use interface for easily integrating differentiable QPs into machine learning algorithms or bi-level programming. Our implementation offers full end-to-end support for both dense and sparse problems with appropriate QP and linear solvers. As a PyTorch module, it is necessary to render Equation (7) as a backpropagation step, which we describe in Appendix C. To ensure modularity, we

offer complete flexibility in selecting a QP solver for the forward pass by interfacing with the open-source *qpsolvers* library (Caron et al., 2024b). Their library provides a minimal-overhead interface supporting over 15 free and commercial QP solvers, and easily supports the integration of additional solvers. We similarly provide flexibility in choosing the linear solver used for differentiation: our code supports several popular direct linear solvers. These include solvers for large-scale sparse systems, like Pardiso (Schenk & Gärtner, 2004), and solvers for symmetric indefinite KKT systems, such as QDLDL (Stellato et al., 2020; Davis, 2005). For users who wish to determine the "best" QP solver for their problem, dQP includes a simple profiling tool (see Appendix D). More details are given in Appendix E including constraint normalization, handling non-differentiable points, and options like warm-starting for bi-level optimization.

# 4 EXPERIMENTAL RESULTS

We have extensively tested dQP to ensure its robustness, evaluate its performance against competing methods for differentiable quadratic programming, and demonstrate its applicability and advantages in large-scale structured problems. Notably, we emphasize dQP's strengths in handling large, sparse problems, complementing custom differentiable GPU-batched solvers such as OptNet (Amos & Kolter, 2017), which are optimized for solving many small, dense problems simultaneously. Given this focus, and considering the limited availability of state-of-the-art GPU-batchable QP solvers, we conduct our experiments on CPUs, similar to prior works such as QPLayer, SCQPTH, and Alt-Diff (Bambade et al., 2024; Butler, 2023; Sun et al., 2022). Our evaluation includes a large benchmark consisting of over 2,000 dense and sparse challenging QPs taken from public datasets as well as randomly generated problems, designed to test dQPs robustness and performance. We also present two prototype applications, demonstrating the applicability of dQP in a learning experiment and in bi-level optimization. See Appendix G for the full details on each experiment's configuration.

**Modularity and Performance.** We tested dQP on nearly 200 QPs from the QP benchmark (Caron et al., 2024a), including 65 small Model Predictive Control (MPC) problems and 129 challenging, sparse problems from the standard Maros-Meszaros (MM) dataset (Maros & Mészáros, 1999), which includes large-scale instances. These problems are designed to serve as stress tests for QP solvers. We compared dQP with other differentiable QP methods: OptNet (Amos & Kolter, 2017), QPLayer (Bambade et al., 2024), and SCQPTH (Butler, 2023), each integrated with its specialized QP solver. For its forward pass, dQP was paired with the leading QP solver for each problem as reported by QP Benchmark (Caron et al., 2024a). The total runtime (forward and backward passes), accuracy (duality gap), and dimension (illustrated by point size) are reported in the scatter plots in Figure 4 for each problem and each differentiable QP solver. The average performance across the entire dataset and on the subset

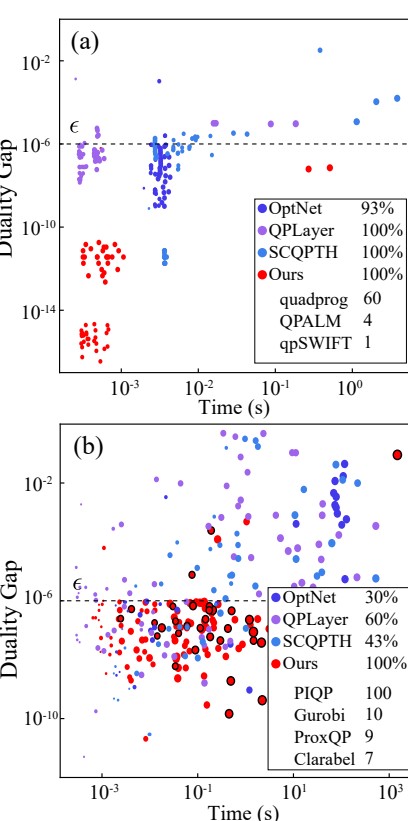

Figure 4: Accuracy versus total forward/backward solve for the (a) MPC and (b) Maros-Meszaros datasets. Each point represents a solved problem; point size illustrates dimension; problems solved solely by dQP circled. Legend shows percentages of success rates, and counts of forward solvers used by dQP for each problem.

of problems solved by all methods is shown in Table 1. For small, dense MPC problems, dQP is typically comparable to QPLayer while achieving much higher accuracy. MM problems, being significantly more challenging, often cause competing methods to fail. OptNet and SCQPTH solved less than 50% of the problems, while dQP successfully solved all MM problems and was the *only* differentiable solver to succeed in 38 of them (circled in the figure). Moreover, dQP was the fastest and most accurate in 81% and 83% of all problems, respectively. It particularly excelled in larger

| Dataset | Solver | Full Dataset | | | | | | Subset of Problems Solved by All Methods | | | | | |
|---|---|---|---|---|---|---|---|---|---|---|---|---|---|
| | | # Probs Solved | Avg Fwd [ms] | Avg Bwd [ms] | Avg Total [ms] | Avg Bwd/Total | Accuracy [duality gap] | # Probs Solved | Avg Fwd [ms] | Avg Bwd [ms] | Avg Total [ms] | Avg Bwd/Total | Accuracy [duality gap] |
| MPC | dQP | **65** | **1.19** | 14.42 | 15.61 | 42% | **$1.15 \times 10^{-8}$** | 60 | 0.30 | 0.19 | 0.50 | 38% | **$1.02 \times 10^{-8}$** |
| | QPLayer | **65** | 4.19 | 0.85 | 5.05 | 41% | $2.17 \times 10^{-5}$ | 60 | **0.23** | **0.16** | **0.39** | 43% | $2.28 \times 10^{-5}$ |
| | OptNet | 60 | 2.82 | **0.30** | **3.12** | **9%** | $1.76 \times 10^{-5}$ | 60 | 2.82 | 0.30 | 3.12 | **9%** | $1.76 \times 10^{-5}$ |
| | SCQPTH | **65** | 134.75 | 2.17 | 136.93 | 11% | $5.02 \times 10^{-4}$ | 60 | 11.97 | 0.49 | 12.46 | 12% | $5.39 \times 10^{-4}$ |
| MM | dQP | 129 | 471 | 996 | **1467** | 57% | **$7.39 \times 10^{-6}$** | 24 | **10** | **83** | **93** | 35% | **$1.73 \times 10^{-7}$** |
| | QPLayer | 77 | 15089 | **632** | 15721 | 18% | $2.21 \times 10^{-2}$ | 24 | 2828 | 433 | 3261 | 29% | $1.77 \times 10^{-4}$ |
| | OptNet | 38 | 39329 | 2139 | 41468 | **6%** | $2.36 \times 10^{-3}$ | 24 | 9199 | 559 | 9758 | **7%** | $1.71 \times 10^{-4}$ |
| | SCQPTH | 55 | 16344 | 6551 | 22895 | 13% | $1.81 \times 10^{-2}$ | 24 | 14048 | 3019 | 17067 | 14% | $8.75 \times 10^{-3}$ |

Table 1: Performance of differentiable QP methods for 65 small Model Predictive Control (MPC) problems and 129 challenging, sparse problems from the Maros-Meszaros (MM) dataset.

problems (dimension over 1000), where it was the fastest and most accurate in 98% and 95% of cases, respectively. Further technical details, along with additional experiments on 450 random dense QPs and 625 sparse QPs with dimensions ranging from 10 to $10^4$, are provided in Appendix G.1.1, showing similar results.

**Scalability.** We evaluated dQP on large-scale sparse problems, a regime where state-of-the-art QP solvers hold a significant advantage over less optimized solvers. We tested dQP and other available differentiable QP solvers on two prototypical projection layers expressed as constrained QPs:

$$P_1(x) = \arg\min_z \|x - z\|_2^2 \text{ subject to } 0 \le z \le 1, \sum z_i = 1, \text{ and}$$

$$P_2(x_1, \dots, x_n) = \arg\min_{z_1, \dots, z_n} \sum \|x_j - z_j\|_2^2 \text{ subject to } \|z_j - z_{j+1}\|_\infty \le 1.$$

Results for $P_1$ are shown in Figure 1, demonstrating dQP's scalability compared to OptNet and QPLayer. Other methods fail on all but small problems (see Appendix G.1.2). In dimensions greater than 2000, dQP outperforms competing methods by 2-3 orders of magnitude in both speed and accuracy. Competing methods are limited to dense calculations and fail in dimensions beyond $10^4$. It's worth noting that $P_1$ is the projection onto the probability simplex, also known as SparseMAX, for which more efficient, non-QP-based methods exist (Martins & Astudillo, 2016). Results for $P_2$, representing projection onto "chains" with bounded links, exhibit similar scalability and are detailed in Appendix G.1.3.

**Learning Sudoku.** We evaluated dQP in a popular learning setting, first introduced in OptNet (Amos & Kolter, 2017). In this experiment, linear constraints model the rules of the Sudoku game, which are then learned from examples of solved Sudoku boards via differentiable QPs. We reproduced the experiment from (Amos & Kolter, 2017) by integrating dQP into their code. As shown in

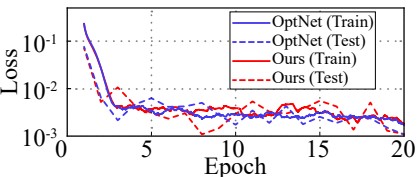

the inset, the training and testing losses achieved by OptNet and dQP paired with PIQP (Schwan et al., 2023) are comparable. This experiment similar to the MPC results shown in Figure 4(a), validates dQP's ability to perform on par with leading differentiable QP packages. However, we note that, compared to tightly integrated forward-backward implementations, dQP has some disadvantages, *e.g.*, QPLayer supports differentiation of infeasible QPs, and OptNet natively supports GPU batching, which is not available for black-box state-of-the-art sparse QP solvers, and thus cannot be easily integrated with dQP.

**Bi-Level Geometry Optimization.** We further test dQP in a non-learning, optimization-based setting inspired by the geometric problem of intersection-free straight-edge planar graph drawing: embed a planar graph representing a triangular mesh into a non-convex domain, such edges are drawn as straight non-overlapping lines. Kovalsky et al. (2020) formulate a linear inequality constraint-satisfaction problem for which they show exists an (unknown) Laplacian $M$ that defines a quadratic energy and thus a QP which, when solved, yields exactly such a straight-edge drawing. However, their conditions are nonconstructive and have remained theoretical. We cast this problem as the following bi-level optimization problem:

$$M^* = \arg\min_M \ \|\mu^*(M)\|_2^2$$

$$\text{subject to } (v^*(M), \lambda^*(M), \mu^*(M)) = \arg\min_v \left\{ \text{tr}\left(v^T M v\right) \text{ subject to } Bv = u, \ CMv \succeq 0 \right\}$$

where $v \in \mathbb{R}^{n \times 2}$ represents the $n$ coordinates of the mesh vertices, $M \in \mathbb{R}^{n \times n}$ is a parameterized Laplacian, and $B, u$ and $C$ encode the boundary conditions of Kovalsky et al. (2020). The results of Kovalsky et al. (2020) then imply that $v^*(\mathcal{L}^*)$ represents a straight line drawing if the dual variable $\mu^*(M)$, corresponding to linear inequalities of the nested optimization, vanishes.

In our experiments, we solve this bi-level problem using dQP paired with PIQP. The inset shows an example of this experiment: (a) the triangulated unit square is the chosen graph; (b) an invalid embedding produced by choosing an arbitrary Laplacian;

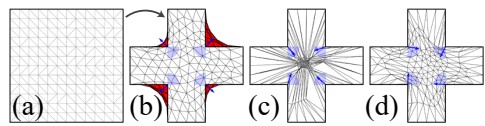

(c) a valid embedding which minimizes the above bi-level problem; (d) with additional regularization on the shape of the triangles.

Figure 5 shows a refinement experiment showing that dQP scales favorably as mesh size increases compared to OptNet, QPLayer and SCQPTH; in particular, only dQP scales up to problems with over $10^4$ vertices. We only report forward (QP) time for OptNet, QPLayer and SCQPTH because OptNet and SCQPTH do not output, nor differentiate the duals, and while QPLayer does, it suffers poor scaling from dense operations as the others. Lastly, Figure 2 presents the large-scale bijective embedding of an ant mesh.

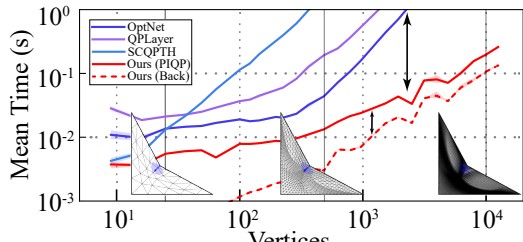

Figure 5: Solver speed under mesh refinement for mapping into a non-convex perturbed square, visualized at different resolutions.

## 5    CONCLUSION

dQP is shown to provide a differentiable interface to any QP solver, and yield an extremely efficient QP-based layer which can be used in, *e.g.*, neural architectures. We believe this work is the first step in providing similar differentiable layers for other popular optimization problems (*e.g.*, semidefinite programming), which we plan to tackle next. Additionally, we note that our current method does not enable neither full parallelization nor GPU support, and we mark these as important challenges to tackle.

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

# A PROOF OF THEOREM 1

In this section we provide a proof of Theorem 1, which we restate below:

**Theorem 1.** *The QP Equation (5) is locally equivalent to the reduced equality-constrained QP Equation (1) and its solution $\zeta^*(\theta) = (z^*(\theta), \lambda^*(\theta), \mu^*(\theta))$ admits the explicit form*

$$\begin{bmatrix} z^* \\ \lambda^* \\ \mu_J^* \end{bmatrix} = \begin{bmatrix} P & A^T & C_J^T \\ A & 0 & 0 \\ C_J & 0 & 0 \end{bmatrix}^{-1} \begin{bmatrix} -q \\ b \\ d_J \end{bmatrix}. \tag{6}$$

*Furthermore, the optimal point can be explicitly differentiated to obtain*

$$\begin{bmatrix} \partial_\theta z^* \\ \partial_\theta \lambda^* \\ \partial_\theta \mu_J^* \end{bmatrix} = - \begin{bmatrix} P & A^T & C_J^T \\ A & 0 & 0 \\ C_J & 0 & 0 \end{bmatrix}^{-1} \left( \begin{bmatrix} \partial_\theta P & \partial_\theta A^T & \partial_\theta C_J^T \\ \partial_\theta A & 0 & 0 \\ \partial_\theta C_J & 0 & 0 \end{bmatrix} \begin{bmatrix} z^* \\ \lambda^* \\ \mu_J^* \end{bmatrix} - \begin{bmatrix} -\partial_\theta q \\ \partial_\theta b \\ \partial_\theta d_J \end{bmatrix} \right). \tag{7}$$

*Proof.* We begin by establishing that the QP Equation (1) and the equality-constrained reduced QP Equation (5) are equivalent. For any $\theta$ satisfying the assumptions of the theorem, the QP Equation (1) has a unique solution characterized by the KKT system

$$\begin{aligned} P(\theta)z^*(\theta) + q(\theta) + A(\theta)^T \lambda^*(\theta) + C(\theta)^T \mu^*(\theta) &= 0 \\ A(\theta)z^*(\theta) - b(\theta) &= 0 \\ C(\theta)z^*(\theta) - d(\theta) &\leq 0 \\ \mu^*(\theta) &\geq 0 \\ D(\mu^*(\theta))(C(\theta)z^*(\theta) - d(\theta)) &= 0. \end{aligned} \tag{8}$$

Complementarity implies that active constraints $j \in J(\theta)$ have $\mu^*(\theta)_j > 0$ and therefore must be satisfied with an equality $(C(\theta)z^*(\theta) - d(\theta))_j = 0$, while inactive constraints $j \notin J(\theta)$ have $\mu^*(\theta)_j = 0$ and thus can be eliminated, without altering the solution. Therefore, the unique solution $\zeta^*(\theta) = (z^*(\theta), \lambda^*(\theta), \mu^*(\theta))$ of Equation (8) is also the unique solution of the reduced system

$$\begin{aligned} P(\theta)z^*(\theta) + q(\theta) + A(\theta)^T \lambda^*(\theta) + C(\theta)_{J(\theta)}^T \mu^*(\theta)_{J(\theta)} &= 0 \\ A(\theta)z^*(\theta) - b(\theta) &= 0 \\ C(\theta)_{J(\theta)} z^*(\theta) - d(\theta)_{J(\theta)} &= 0, \end{aligned} \tag{9}$$

which are exactly the KKT conditions of the equality-constrained reduced QP Equation (5). Uniqueness of solution then implies that Equation (1) and Equation (5) are pointwise equivalent at $\theta$. Moreover, since $P, q, A, b, C, d$ are smoothly parameterized by $\theta$, the Basic Sensitivity Theorem (Fiacco, 1976) asserts that the primal-dual solution $\zeta^*(\theta)$ for Equation (1) is a differentiable function of $\theta$ in a neighborhood of $\theta$, defined implicitly through the KKT's equality conditions. Furthermore, the active set $J(\theta)$ is fixed in this neighborhood, therefore Equation (1) and Equation (5) are locally equivalent.

Equation (9) implies that the reduced primal-dual solution $\zeta_J^*(\theta) = (z^*(\theta), \lambda^*(\theta), \mu_J^*(\theta))$ satisfies $K_J(\theta)\zeta_J^*(\theta) = v_J(\theta)$, where

$$K_J(\theta) = \begin{bmatrix} P(\theta) & A(\theta)^T & C(\theta)_{J(\theta)}^T \\ A(\theta) & 0 & 0 \\ C(\theta)_{J(\theta)} & 0 & 0 \end{bmatrix}, \quad v_J(\theta) = \begin{bmatrix} -q(\theta) \\ b(\theta) \\ d(\theta)_{J(\theta)} \end{bmatrix}. \tag{10}$$

Under the assumptions of the theorem, the reduced KKT matrix $K_J(\theta)$ is invertible and

$$\zeta_J^* = K_J^{-1} v_J, \tag{11}$$

yielding Equation (6). Moreover, since $J(\theta)$ is locally constant, the Basic Sensitivity Theorem establishes that $\zeta_J^*(\theta)$ is differentiable. Using the formal derivative of the matrix inverse (Magnus & Neudecker, 1988; Petersen & Pedersen, 2008) we *explicitly* differentiate Equation (11) to obtain

$$\partial_\theta \zeta_J^* = (-K_J^{-1}(\partial_\theta K)K_J^{-1})v_J + K_J^{-1}(\partial_\theta v_J) = -K_J^{-1}(\partial_\theta K_J)\zeta_J^* + K_J^{-1}(\partial_\theta v_J), \tag{12}$$

yielding Equation (7).

$\square$

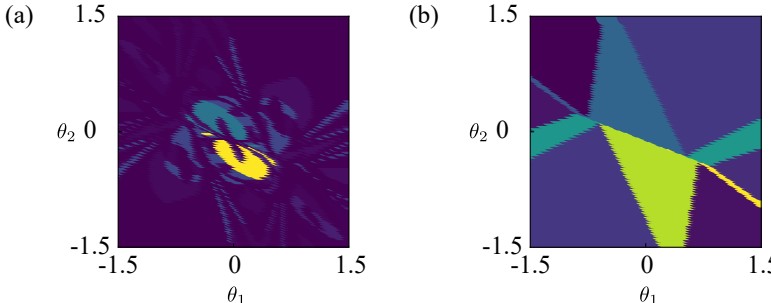

Figure 6: The set-up in figure 3.3 with looser solver tolerance $\epsilon_{\text{abs}} = 10^{-4}$, active tolerance $\epsilon_J = 10^{-7}$, and solver PIQP. (a) The computed active set is degraded due to the inaccurate solution. (b) Our heuristic active set refinement algorithm recovers the ground truth active sets.

## B  ACTIVE SET REFINEMENT

Inaccuracy in a solution may lead to instability in the active set near weakly active constraints, degrading the gradient quality. To show this, we repeat the experiment in Figure 3.3 which has a simple polyhedral active set parameter space. One setup where instability appears is illustrated in Figure 6 where we use absolute solver tolerance $\epsilon_{\text{abs}} = 10^{-4}$ and active tolerance $\epsilon_J = 10^{-7}$. Qualitatively, the active set at each solution is severely degraded, even for points away from the boundaries where the set changes. We provide a *optional* heuristic algorithm to address this, which recovers the desired set in this problem. First, we order the constraints by increasing residual and select an initial active set from the tolerance $\epsilon_J$. Then, we progressively add constraints by checking if the residual of the system 6 for $\zeta_J^*$ decreases, and greedily accepting until adding constraints no longer improves the residual. At each step, we keep the primal solution from the forward fixed, and solve for the new active dual variables. While this algorithm works well on simple examples, more sophisticated and efficient techniques may be desired for harder problems. We did not use this refinement algorithm in any of our experiments.

## C  BACKPROPAGATION

Like other differentiable QP layers implemented within automatic differentiation frameworks such as PyTorch (Paszke et al., 2019), we do not directly compute the derivative $\partial_\theta \zeta^*$. Specifically, dQP directly receives the QP parameters $P, q, A, b, C, d$ and not $\theta$, and so in backpropagation we are not concerned with $\theta$. This is rather accounted for in the next step outside dQP, usually by automatic differentiation. Instead, backpropagation requires that we compute a so-called Jacobian-vector product which are products of the Jacobians with an "incoming" gradient of a quantity or loss $\ell$ that depends on $\zeta^*$. This requires less computation and does not require the formation of a 3-tensor. Since $\zeta_J^* = K_J^{-1} v_J$ is a formal matrix-vector multiplication, the Jacobian-vector product is well-known,

$$\nabla_{v_J} \ell = (K_J^{-1})^T \, \nabla_{\zeta_J^*} \ell, \tag{13}$$

and

$$\nabla_{K_J} \ell = -\nabla_{v_J} \ell \, \zeta_J^{*T}, \tag{14}$$

with respect to $K_J, v_J$, respectively. Although backpropagation introduces a transposition, the re-use of a factorization from solving for the active duals is unaffected. This follows from the symmetry of the reduced KKT matrix which simplifies Equation (13) into $\nabla_{v_J} \ell = K_J^{-1} \nabla_{\zeta_J^*} \ell$. Next, we extract the gradients with respect to the parameters by the chain rule. This amounts to tracking their position in the blocks and accounting for symmetry constraints. It is helpful to write $(d_z, d_\lambda, d_{\mu_J}) = -\nabla_{v_J} \ell$ so that we express

$$
\begin{aligned}
\nabla_P \ell &= \frac{1}{2} \left( d_z z^{*T} + z^* d_z^T \right) & \nabla_q \ell &= d_z \\
\nabla_A \ell &= d_\lambda z^{*T} + \lambda^* d_z^T & \nabla_b \ell &= -d_\lambda \\
\nabla_{C_J} \ell &= d_{\mu_J} z^{*T} + \mu_J^* d_z^T & \nabla_{d_J} \ell &= -d_{\mu_J},
\end{aligned}
\tag{15}
$$

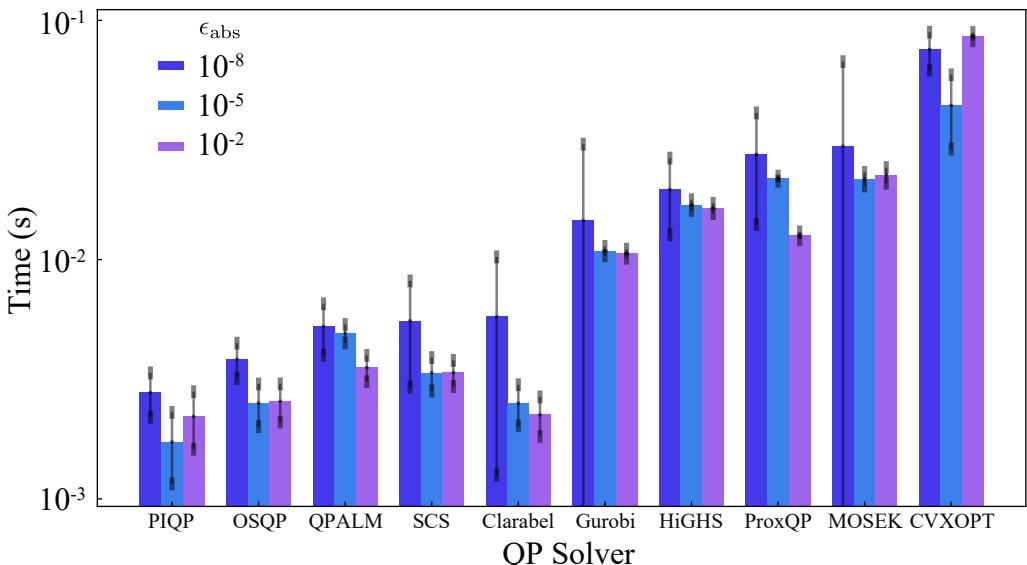

Figure 7: Evaluating the best QP solver for the cross geometry problem using our diagnostic tool. The solution tolerance regimes are varied between $\epsilon_{\text{abs}} = 10^{-8}, 10^{-5}, 10^{-2}$.

similar to OptNet (Amos & Kolter, 2017). We note that the gradient with respect to $P$ is constrained to lie within the subspace of symmetric matrices. Similarly, if the matrices $P, A, C$ are sparse, then we project the gradient to lie within the non-zero entries, which can be implemented efficiently in Equations 15. Although the above argument is for a scalar loss $\ell$, the same approach is naturally adapted if $\zeta^*$ is mapped to a vector in the immediate next layer.

## D   CHOOSING A SOLVER

Since our work enables users to choose any QP solver as the front-end for their differentiable QP applications, we include a simple diagnostic tool for quantitatively measuring solver performance. We present an example result in Figure 7 for the cross geometry experiment in section 4, finding PIQP, OSQP, and QPALM to be the most efficient. For this reason, we choose PIQP in the geometry experiments. We also include tools for checking the solution and gradient accuracy.

## E   IMPLEMENTATION DETAILS

**Tolerances** In addition to the active set tolerance $\epsilon_J$, QP solvers often support additional user-provided tolerances. These include the primal residual which measures violations of feasibility, the dual residual which measures violations of stationary, and for some solvers also the duality gap, which provides a direct handle on solution accuracy. We inherit the structure of *qpsolvers* for setting custom tolerances on different QP solvers, though we set a heuristic default which is sufficient for many of the experiments in this work.

**Convexity and Feasibility** Two key assumptions of our method are strict convexity and feasibility. However, these are often violated in practice. We include optional checks that $P$ is symmetric positive definite. On the other hand, we do not perform any special handling for infeasibility – a limitation of our method compared to, for example, QPLayer (Bambade et al., 2024).

**Non-differentiable Points** For non-differentiable problems, we solve for the derivatives in the least-squares sense, plugging the system into *qpsolvers* which can handle least-squares, or a standard least-squares solver. We attempt to anticipate weakly active constraints which cause non-differentiability by measuring the norms of the primal residual and the dual. The reduced KKT is also non-invertible if the active dual solution is not unique. To detect this, we check a necessary condition: the total number of active constraints plus the number of equalities must be less than the

dimension. Otherwise, if these necessary checks are passed, we attempt the standard linear solve and pass to least-squares if it fails.

**Normalization** Some problems have large variations in scale between different rows within the constraints. This influences the primal residual and thus the active set, which is determined by comparing with an absolute threshold tolerance. To address this issue for these problems, we include an *optional* differentiable normalization step on the constraints before Algorithm 1 is carried out. Under this choice, the resulting relative primal residual becomes the scale-invariant distance to the constraint.

**Equality Constraints** While we include equality constraints in our general formulation, they are not required.

**Warm-Start** Since *qpsolvers* supports warm-starting, we inherit it as an option and store data in the PyTorch module from previous outer iterations, which can be used as initialization. This is useful for bi-level optimization problems where the input $\theta$ changes little between outer iterations.

**Fixed Parameters** For fixed parameters, we do not compute the corresponding derivative. This saves the cost of unwrapping the linear solve as in Equation (15) and saves the memory to form the loss gradients, which are matrices for $P, A, C$.

**Active Set Refinement** See the discussion in Appendix B.

**QP Solvers** Throughout this work, we use a number of QP solvers available in *qpsolvers* including Clarabel (Goulart & Chen, 2024), DAQP (Arnström et al., 2022), Gurobi (Gurobi Optimization, LLC, 2024), HiGHS (Huangfu & Hall, 2018), HPIPM (Frison & Diehl, 2020), MOSEK (Andersen & Andersen, 2000), OSQP (Stellato et al., 2020), PIQP (Schwan et al., 2023), ProxQP (Bambade et al., 2023), QPALM (Hermans et al., 2019), qpSWIFT (Pandala et al., 2019), quadprog (Goldfarb & Idnani, 1983), and SCS (O'Donoghue et al., 2016).

## F VALIDATING ALGORITHM 1

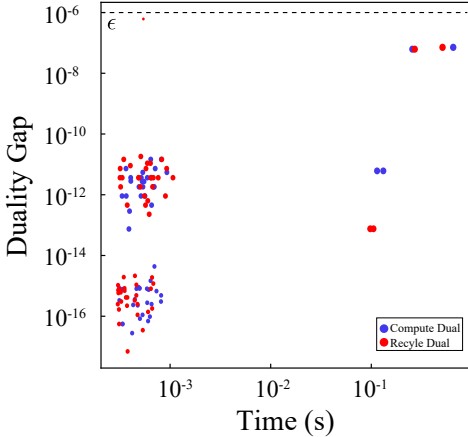

Figure 8: Forward and backward evaluation on the MPC dataset with the dual output by the QP solver (red) and the one obtained by solving 6.

The solvers currently available in *qpsolvers* provide dual solutions. Thus, to validate our modular algorithm which does not require them, we repeat the experiment in Figure 4(a). We ignore the dual solution received from the forward solver, and instead perform the optional step of computing them from the reduced KKT system 6. The results are shown in Figure 8. The additional computation of the duals has a small effect on the total backward time, as we prefactorize $K_J$ and use it for the derivatives as well. Using the reduced KKT to solve for the duals also impacts the duality gap, which can be seen for the larger problems in the MPC dataset, but still respect the absolute tolerance set on the duality gap.

# G  Experimental Details

For completeness and reproducibility, we include additional details on the experiments. We run all experiments and methods on CPU, including methods that support GPU such as OptNet.

## G.1  Performance Evaluation

All experiments in this section were run on a Macbook Air with Apple M2 chips, 8 cores, and 16GB RAM.

In our QP benchmark experiments, we evaluate the solution accuracy using the primal residual $r_p$ (the maximum error on equality and inequality constraints), dual residual $r_d$ (the maximum error on the dual feasibility condition), and duality gap $r_g$ (the difference between primal and dual optimal values).

$$r_p = \max\left(\|Az - b\|_\infty, [Cz - d]_+\right)$$
$$r_d = \|Pz + q + A^T\lambda + C^T\mu\|_\infty$$
$$r_g = |z^T Pz + q^T z + b^T\lambda + d^T\mu|$$

Throughout our experiments, we present results for the duality gap to indicate the solution accuracy since, for a strongly convex QP, a zero duality gap $r_g = 0$ is a necessary and sufficient condition for optimality.

For the forward, we set the absolute residual tolerance to $10^{-6}$. We set the active constraint tolerance to $\epsilon_J = 10^{-5}$. We run each problem separately with batch size 1.

In our benchmark, we regard a problem as successfully solved if it meets the following criteria:

1. The solve time is less than a practical 800s time limit.

2. The primal residual, dual residual, and duality gap are less than 1.0. This is a coarse check, less stringent than the imposed tolerances.

3. The differentiation is executed, and does not lead to a fatal error (e.g. due to non-invertibility of a linear system).

Experimental results are averaged over 5 independent samples.

Since SCQPTH does not support equality constraints, we convert them into an corresponding set of inequality constraints.

### G.1.1  Random dense/sparse problems

We generated two sets of random QPs: dense and sparse. For the dense set, the data is generated as $P = Q^T Q + 10^{-4}I$, where $Q \in \mathbb{R}^{n \times n}$ with $Q_{ij} \sim \mathcal{U}(0, 1)$, $C \in \mathbb{R}^{m \times n}$ with $C_{ij} \sim \mathcal{U}(0, 1)$, $d = C\mathbb{1} + \mathbb{1}$, and $A \in \mathbb{R}^{p \times n}$ with $A_{ij} \sim \mathcal{U}(0, 1)$, $b = A\mathbb{1}$. The set, with 450 problems, has dimensions $n \in \{10, 20, 50, 100, 220, 450, 1000, 2100, 4600\}$, $m = n$, and $p = n/2$. Each dimension contains 50 problems. We use DAQP and ProxQP as the forward solvers. Figure 9 and Table 2 show that our method is comparable to OptNet and QPLayer in both time and accuracy. For smaller dimensions ($n \leq 1000$), DAQP provides higher accuracy, while for larger problems, ProxQP is more efficient.

For the sparse set, $P$ is generated as $P = L^T L$, where $L$ is the standard Laplacian matrix of $k$-nearest graph ($k = 3$). Entries of $C$ and $A$ are filled by $\mathcal{N}(0, 1)$ random numbers with density of $5 \times 10^{-4}$ and zero row is avoided. The vectors $d$ and $b$ are generated similarly to the dense set. The set, with 625 problems, has dimensions $n \in \{100, 220, 450, 1000, 2100, 4600, 10000\}$, with $m = n$ and $p = n/2$. For $n \leq 4600$, each dimension contains 100 problems and 25 problems for $n > 4600$. KKT systems in these problems tend to be ill-conditioned. We use Gurobi as the forward solver and employ least squares solver for backward. In our experiments OptNet fails on all problems and SCQPTH is substantially slower and fails for $n \geq 4600$, and are thus excluded in our report. Figure 9 and Table 3 demonstrate our superior accuracy and efficiency over QPLayer.

| Solver | Metric | Problem Size | | | | | |
|---|---|---|---|---|---|---|---|
| | | 20 | 100 | 450 | 1000 | 2100 | 4600 |
| dQP (daqp) | Accuracy | $\mathbf{1.59 \times 10^{-11}}$ | $\mathbf{1.20 \times 10^{-8}}$ | $2.35 \times 10^{-6}$ | $4.08 \times 10^{-5}$ | $5.26 \times 10^{-4}$ | Failed |
| | Forward [ms] | 0.20 | 1.31 | 131.35 | 1115.62 | 10065.77 | - |
| | Backward [ms] | **0.14** | 0.48 | 11.22 | 56.90 | 313.90 | - |
| | Total [ms] | 0.34 | 1.81 | 144.91 | 1174.47 | 10379.68 | - |
| dQP (proxqp) | Accuracy | $4.71 \times 10^{-6}$ | $6.42 \times 10^{-5}$ | $9.11 \times 10^{-4}$ | $7.26 \times 10^{-4}$ | $4.13 \times 10^{-4}$ | $4.25 \times 10^{-4}$ |
| | Forward [ms] | 0.29 | 2.54 | 61.12 | **379.74** | **2553.82** | **26408.12** |
| | Backward [ms] | 0.17 | 1.85 | 13.53 | 70.25 | 385.04 | 3369.77 |
| | Total [ms] | 0.46 | 4.32 | 73.68 | **455.22** | **2935.93** | **29771.33** |
| OptNet | Accuracy | $6.89 \times 10^{-8}$ | $2.51 \times 10^{-8}$ | $\mathbf{3.80 \times 10^{-8}}$ | $\mathbf{3.51 \times 10^{-7}}$ | $\mathbf{2.80 \times 10^{-6}}$ | $\mathbf{3.34 \times 10^{-5}}$ |
| | Forward [ms] | 2.99 | 7.09 | 78.56 | 463.04 | 3176.59 | 29387.34 |
| | Backward [ms] | 0.23 | 0.45 | **5.55** | **29.30** | **185.80** | **1540.07** |
| | Total [ms] | 3.22 | 7.56 | 84.20 | 491.57 | 3362.25 | 30931.00 |
| QPLayer | Accuracy | $3.08 \times 10^{-6}$ | $6.88 \times 10^{-5}$ | $3.98 \times 10^{-5}$ | $1.31 \times 10^{-4}$ | $1.35 \times 10^{-5}$ | $1.48 \times 10^{-4}$ |
| | Forward [ms] | **0.14** | **0.99** | **43.11** | 407.77 | 3973.89 | 43740.91 |
| | Backward [ms] | 0.15 | **0.34** | 9.67 | 74.24 | 601.25 | 5781.58 |
| | Total [ms] | **0.29** | **1.35** | **52.99** | 482.17 | 4575.13 | 49558.44 |
| SCQPTH | Accuracy | $3.48 \times 10^{-5}$ | $4.62 \times 10^{-4}$ | $4.32 \times 10^{-5}$ | $6.54 \times 10^{-5}$ | $1.83 \times 10^{-4}$ | $2.26 \times 10^{-3}$ |
| | Forward [ms] | 10.01 | 26.72 | 120.12 | 664.74 | 6802.36 | 384565.40 |
| | Backward [ms] | 0.47 | 1.28 | 26.80 | 184.22 | 1733.72 | 15203.05 |
| | Total [ms] | 10.50 | 27.90 | 147.25 | 850.37 | 8550.04 | 399699.87 |

Table 2: Time and accuracy performance statistics on random dense problems.

| Solver | Metric | Problem Size | | | | | | |
|---|---|---|---|---|---|---|---|---|
| | | 100 | 220 | 450 | 1000 | 2100 | 4600 | 10000 |
| dQP (Gurobi) | **Accuracy** | $\mathbf{4.46 \times 10^{-8}}$ | $\mathbf{9.23 \times 10^{-8}}$ | $\mathbf{1.34 \times 10^{-7}}$ | $\mathbf{6.89 \times 10^{-7}}$ | $\mathbf{1.34 \times 10^{-6}}$ | $\mathbf{3.16 \times 10^{-6}}$ | $\mathbf{3.43 \times 10^{-6}}$ |
| | Forward [ms] | 2.57 | **3.44** | **5.53** | **11.07** | **60.68** | **2446.70** | **143209.89** |
| | Backward [ms] | 1.79 | 2.86 | **4.73** | **9.70** | **24.03** | **309.10** | **9364.61** |
| | Total [ms] | 4.37 | **6.33** | **10.28** | **20.72** | **90.01** | **2760.07** | **151471.27** |
| QPLayer | Accuracy | $6.46 \times 10^{-6}$ | $1.25 \times 10^{-5}$ | $1.69 \times 10^{-5}$ | $3.04 \times 10^{-5}$ | $6.12 \times 10^{-5}$ | $1.77 \times 10^{-3}$ | $7.82 \times 10^{-5}$ |
| | Forward [ms] | 1.04 | 5.47 | 31.11 | 235.00 | 2268.24 | 23597.22 | 199009.91 |
| | Backward [ms] | **0.30** | **1.17** | 7.46 | 51.00 | 393.68 | 3538.53 | 38466.29 |
| | Total [ms] | **1.34** | 6.63 | 38.56 | 285.99 | 2658.82 | 27133.19 | 240084.62 |

Table 3: Time and accuracy performance statistics on random sparse problems.

### G.1.2 PROJECTION ONTO THE PROBABILITY SIMPLEX

This formulation projects a vector onto the probability simplex, as formulated in $P_1$. We set $x \in \mathbb{R}^n$ with $x_i \sim \mathcal{N}(0,1)$. The set, with 500 problems, has dimensions $n \in \{10, 20, 50, 100, 220, 450, 1000, 2100, 4600, 10000, 100000\}$. For $n \leq 4600$, each dimension contains 50 problems and 25 problems for $n > 4600$. Gurobi serves as our forward sparse solver. Figure 1 shows the median performance within the $1/4$ and $3/4$ quantiles for each dimension. SCQPTH failed for all problems with $n > 50$ is is thus excluded from our report. The statistics in Table 4 show that we outperform competing methods for differentiable QP in both forward and backward times.

### G.1.3 PROJECTION ONTO CHAIN

As formulated in $P_2$, this experiment projects the input point cloud $x_1, ..., x_m \in \mathbb{R}^d$ onto a chain with link of length bounded by 1 in $\infty$-norm. We set $x_i \sim \mathcal{N}(0, 100 I_d)$, with the number of points $m = 100$. By varying the dimension of the vector, $d$, we generated 300 problems in this set with dimensions $n \in \{200, 500, 1000, 2000, 4000, 10000, 100000\}$. For $n \leq 4000$, each dimension contains 50 problems and 25 problems for $n > 4000$. Gurobi was used as our forward solver. Figure 10 and Table 5 show performance similar to that shown in Figure 1 in terms of efficiency. In addition, dQP successfully solves large-scale problems other solvers fail to solve.

### G.2 SUDOKU

The Sudoku experiment was run on an Intel(R) Core(TM) i7-8850H CPU @ 2.60GHz with 6 cores and 16GB RAM.

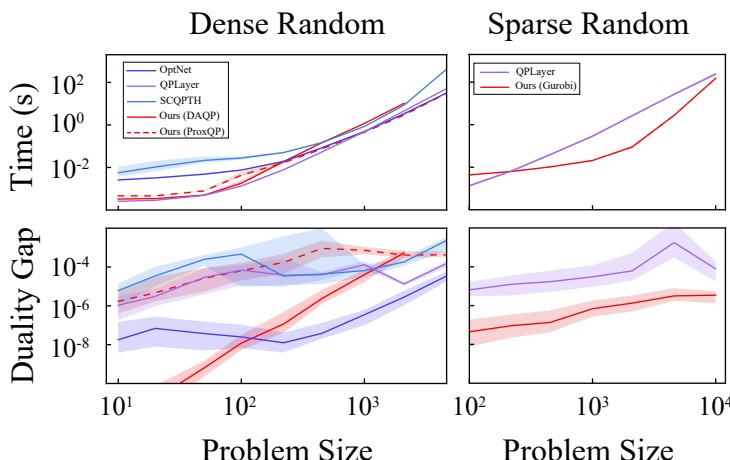

Figure 9: Time and accuracy performance on random dense/sparse problems.

| Solver | Metric | Problem Size | | | | | | |
|--------|--------|------|------|------|------|------|------|--------|
| | | 20 | 100 | 450 | 1000 | 4600 | 10000 | 100000 |
| dQP (Gurobi) | Accuracy | $1.07 \times 10^{-9}$ | $8.88 \times 10^{-10}$ | $2.26 \times 10^{-9}$ | $1.47 \times 10^{-9}$ | $2.72 \times 10^{-9}$ | $9.55 \times 10^{-10}$ | $6.67 \times 10^{-10}$ |
| | Forward [ms] | 1.38 | 1.65 | **2.66** | 4.37 | 15.83 | 42.21 | 423.91 |
| | Backward [ms] | 0.24 | **0.28** | **0.46** | 0.69 | 2.58 | 6.21 | 53.45 |
| | Total [ms] | 1.63 | 1.92 | **3.13** | **5.06** | **18.46** | **49.00** | **476.64** |
| OptNet | Accuracy | $4.04 \times 10^{-8}$ | $4.24 \times 10^{-8}$ | $1.64 \times 10^{-8}$ | $2.67 \times 10^{-8}$ | $3.95 \times 10^{-8}$ | $6.08 \times 10^{-8}$ | Failed |
| | Forward [ms] | 2.72 | 4.72 | 33.46 | 165.50 | 7788.73 | 65976.45 | – |
| | Backward [ms] | 0.20 | 0.46 | 3.99 | 17.48 | 720.43 | 4958.74 | – |
| | Total [ms] | 2.92 | 5.19 | 37.66 | 182.99 | 8514.65 | 70856.43 | – |
| QPLayer | Accuracy | $9.53 \times 10^{-6}$ | $3.65 \times 10^{-5}$ | $4.16 \times 10^{-4}$ | $2.19 \times 10^{-4}$ | $1.16 \times 10^{-3}$ | $1.94 \times 10^{-3}$ | Failed |
| | Forward [ms] | **0.14** | **1.23** | 66.73 | 657.88 | 71724.25 | 869532.53 | – |
| | Backward [ms] | **0.14** | 0.37 | 10.85 | 91.72 | 7594.93 | 77831.58 | – |
| | Total [ms] | **0.29** | **1.61** | 77.56 | 751.14 | 79314.49 | 946174.68 | – |

Table 4: Time and accuracy performance statistics for projection onto the probability simplex.

The set-up of the Sudoku problem is a perturbed linear program

$$z^*(q; A, b) = \arg\min_z \quad \alpha z^T z + q^T z$$
$$\text{subject to} \quad Az = b \tag{16}$$
$$z \geq 0,$$

where $q$ encodes the input unsolved board and $z^*(q)$ encodes the solved board. We distinguish $q$ from the other input data, the constraints $A, b$, which model the Sudoku rules are *learnable* parameters that are optimized by minimizing the mean squared error to the ground truth solution for training boards. Instead of treating $A, b$ as completely independent, they are parameterized to ensure feasibility. The perturbation $\alpha = 0.1$ makes the problem amenable to differentiable quadratic programming.

Our reproduction of the 2x2 Sudoku experiment from OptNet follows closely with their original settings (Amos & Kolter, 2017), except that we use batch size one, run exclusively on CPU, and modify the solution tolerances. For OptNet and dQP, we use the same solution tolerance $\epsilon_{abs} = 10^{-6}$, and for dQP, we use the active tolerance $\epsilon_J = 10^{-5}$. We use the optimizer Adam with learning rate $10^{-3}$ for both methods for the training over 10000 samples, split into 9000 training and 1000 test samples (Kingma & Ba, 2017).

### G.3 BI-LEVEL GEOMETRY OPTIMIZATION

The geometry experiments were run on an Intel(R) Core(TM) i7-8850H CPU @ 2.60GHz with 6 cores.

The cross and ant (Figure 2) meshes and boundary constraints are obtained from the datasets in (Du et al., 2020). We create the mesh refinement example in Figure 5 by perturbing the corner

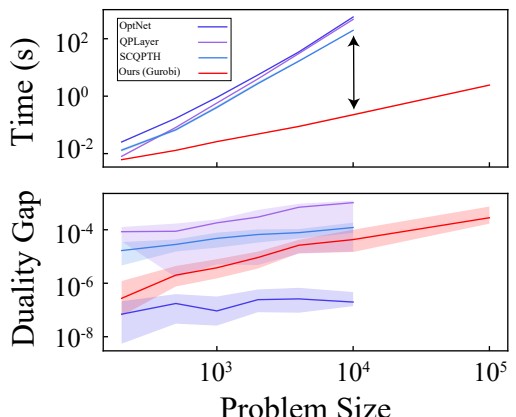

Figure 10: Time and accuracy performance for projection onto chains.

| Solver | Metric | Problem Size | | | | | | |
|---|---|---|---|---|---|---|---|---|
| | | 200 | 500 | 1000 | 2000 | 4000 | 10000 | 100000 |
| dQP (Gurobi) | Accuracy | $2.73 \times 10^{-7}$ | $2.02 \times 10^{-6}$ | $3.79 \times 10^{-6}$ | $9.16 \times 10^{-6}$ | $2.64 \times 10^{-5}$ | $4.29 \times 10^{-5}$ | $\mathbf{2.81 \times 10^{-4}}$ |
| | Forward [ms] | **5.66** | **12.04** | **24.41** | **44.79** | **82.57** | **209.79** | **2263.54** |
| | Backward [ms] | **0.49** | **0.98** | **1.74** | **3.18** | **5.81** | **14.69** | **172.80** |
| | Total [ms] | **6.15** | **12.99** | **26.19** | **47.94** | **88.35** | **224.89** | **2432.64** |
| OptNet | Accuracy | $\mathbf{6.97 \times 10^{-8}}$ | $\mathbf{1.75 \times 10^{-7}}$ | $\mathbf{9.22 \times 10^{-8}}$ | $\mathbf{2.43 \times 10^{-7}}$ | $\mathbf{2.60 \times 10^{-7}}$ | $\mathbf{1.98 \times 10^{-7}}$ | Failed |
| | Forward [ms] | 23.37 | 156.49 | 845.24 | 5124.87 | 32528.54 | 536702.00 | – |
| | Backward [ms] | 1.98 | 13.06 | 61.41 | 365.02 | 2266.20 | 35438.33 | – |
| | Total [ms] | 25.38 | 169.64 | 907.25 | 5491.56 | 34799.98 | 571710.06 | – |
| QPLayer | Accuracy | $8.46 \times 10^{-5}$ | $8.78 \times 10^{-5}$ | $1.82 \times 10^{-4}$ | $2.97 \times 10^{-4}$ | $6.95 \times 10^{-4}$ | $1.03 \times 10^{-3}$ | Failed |
| | Forward [ms] | 6.60 | 69.90 | 505.05 | 3484.47 | 26921.57 | 414295.22 | – |
| | Backward [ms] | 1.44 | 12.10 | 72.13 | 512.95 | 3833.25 | 57219.68 | – |
| | Total [ms] | 8.04 | 81.93 | 577.11 | 3996.92 | 30748.67 | 471649.91 | – |
| SCQPTH | Accuracy | $1.67 \times 10^{-5}$ | $2.83 \times 10^{-5}$ | $4.76 \times 10^{-5}$ | $6.64 \times 10^{-5}$ | $7.80 \times 10^{-5}$ | $1.21 \times 10^{-4}$ | Failed |
| | Forward [ms] | 10.02 | 39.49 | 236.61 | 1617.88 | 8258.89 | 65507.05 | – |
| | Backward [ms] | 3.17 | 28.46 | 170.88 | 1126.46 | 8374.37 | 129385.97 | – |
| | Total [ms] | 13.20 | 67.55 | 407.13 | 2755.28 | 16628.15 | 195462.18 | – |

Table 5: Time and accuracy performance statistics for projection onto chains.

of a square mesh. Importantly, all of the boundary maps selected in our experiments are free of self-intersections, so that preventing triangle inversions implies the global bijectivity of the maps.

To optimize over Laplacians $M$, we directly parameterize the space of Laplacians; we impose that the diagonals are the absolute row sums during optimization and that the off-diagonals are negative. We also constrain $M$ to have the same sparsity pattern as the combinatorial Laplacian $M_c$. We note that the original conditions of (Kovalsky et al., 2020) were formulated in terms of negative semi-definite Laplacians, and so we must transform the problem into the standard form 1. Since the Laplacian $M$ which takes the place of the quadratic term in 1 has a trivial eigenvalue, the resulting QP does not have strict convexity. To address this, we perturb $M$ by a small scaling of the identity $10^{-4}I$.

Throughout the geometry experiments, we use the same solution tolerance $\epsilon_{abs} = 10^{-5}$ and active tolerance $\epsilon_J = 10^{-4}$ with the forward solver PIQP as determined in Appendix D. For the outer optimization, we use the optimizer Adam with learning rate $10^{-2}$ (Kingma & Ba, 2017). We initialize the bi-level optimization with $M_c$. The optimization for the cross experiment is shown in Figure 11(a) where the unregularized loss is driven to the desired tolerance, accompanied by sudden changes in the active set as the dual variables are driven to zero. We terminate the optimization at convergence, once all constraints are inactive to guarantee a bijective map. For the regularized optimization (Figure 11(b)), we penalize deviations from the initial combinatorial Laplacian up to a rescaling using the regularization $\lambda \|\frac{M}{\|M\|_F} - \frac{M_c}{\|M_c\|_F}\|_\infty$. In the cross shown in section 4, we choose the regularization hyper-parameter to be $\lambda = 10$ after sample testing. This regularization is initially weak and so the duals are driven down, eventually crossing the regularization loss as it increases. Yet, while this slows convergence, it does not prevent it – crucial to reach a bijective map because the conditions in (Kovalsky et al., 2020) require all of the inequality constraints to be inactive.

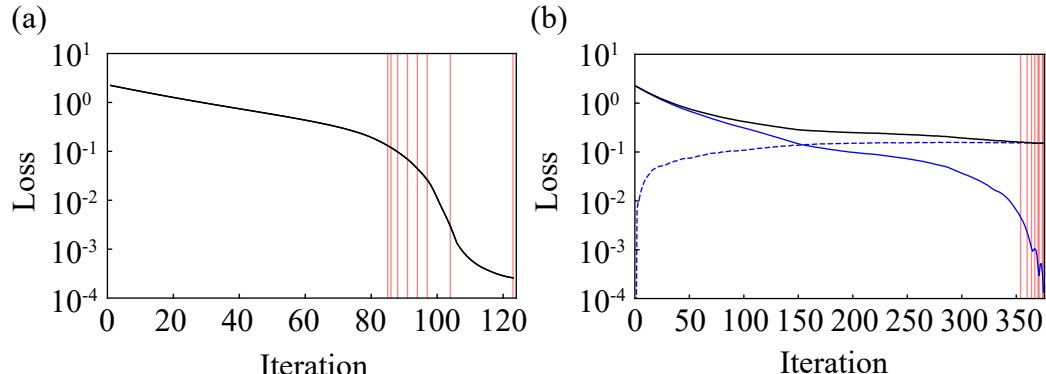

Figure 11: The evolution of the loss for the mappings of the square into the cross. Iterations for which the active set change are denoted with vertical red lines. (a) Without regularization, the loss is driven monotonically to the tolerance. (b) With a competing regularizing loss term (dashed) convergence to the tolerance is slowed but not prevented.

The backward timing that we report in Figure 5 is for the backpropogation through only dQP, as described in Appendix 15. Thus, we remove any contribution coming from the set-up of the parameterized Laplacian and directly report the time to solve the reduced KKT and extract the gradients with respect to $M$.

