# OpenReview forum: "Differentiation Through Black-Box Quadratic Programming Solvers"
_ICLR.cc/2025/Conference — Submitted to ICLR 2025_

### Official Review · Reviewer_odJJ · 2024-10-27

**Soundness:** 2
**Presentation:** 2
**Contribution:** 2
**Rating:** 3
**Confidence:** 5

**Summary:**

This paper presents dQP, a framework for differentiating QP solvers, and conducted the experiments through blackbox solvers for comparing with many existing solvers.The authors demonstrate that dQP outperforms existing models across a range of benchmark problems.

**Strengths:**

The authors present a solid idea aimed at improving the efficiency of the backward process, making it especially effective for large-scale problems.

They conducted numerous experiments and provided a theoretical proof to support their methods.

**Weaknesses:**

I believe the authors do not clearly address their contribution. If the goal of this paper is to propose a generic framework, then it should be clarified how dQP incorporates other solvers. On the other hand, if dQP is intended as a specific method of differentiate opt layer, it should be compared to various state-of-the-art approaches to demonstrate its advantages.


From my understanding, dQP seems to be a novel method for differentiating optimization problems. In Eq (10), it seems all the inequality constraints are tightened into equality constraints. How does this approach ensure the feasibility of the problem? Wouldn’t this compromise the feasibility, potentially resulting in the optimal solution of the modified optimization problem being different from the original one? If that's the case, this framework may not be applicable to predict-then-optimize problems because the optimal solution of the optimization problem has been altered.

However, if this is not the case and a black-box solver is used to solve the problem, then essentially a new forward process is designed for backward optimization but is not actually used, there will be two forward processes, which makes this step seem redundant. Additionally, existing solvers such as OptNet don’t seem to encounter this issue. If this is the main contribution of the paper, I think this it may not be sufficient.


Besides, I’m concerned about the practical performance of dQP (maybe a inexact gradient approach?) I would like to see experimental results comparing dQP with many other existing methods, especially from the aspect of computational efficiency. In the current manuscript, only graphical results are provided, such as in Figure 5, where for around 100 nodes, the results all fall within 1 second, showing minimal differences. Including more numerical results, such as the similarity of gradients computed by different solvers, running times, and differences in the optimal solutions of the optimization problems, and so on, could make the results more convincing.


One more issue is that, in the abstract (and line 355), the authors wrote, "Our implementation, which will be made publicly available, interfaces with an existing framework that supports over 15 state-of-the-art QP solvers." Maybe I think the implementation hasn’t been fully prepared before submission, so the paper may not be fully ready for publication, as it seems there is still some unfinished work.

**Questions:**

1. What are the benefits of dQP over OptNet? Is it more efficient? If so, why? Intuitively, it seems that compared to existing methods, the number of computation steps hasn’t decreased—rather, additional steps related to tolerance testing have been introduced. For example, line 5 in Algorithm 1 appears more complicated.

2. How the tolerance $\epsilon_J$ is set? Couldn’t this lead to infeasibility? I believe the authors should analyze the impact of this parameter in the experiments.

3. Regarding the duality gap, why does OptNet show a larger duality gap in Figure 4, even though the forward method is the same for both OptNet and dQP?

4. The Bi-Level Geometry Optimization in the last part of the experiment is not a common setup. I’m curious about how it can be formulated into KKT conditions. Could you provide more details on this?

5. How is Sparse QP defined in the paper, and what type of problems qualify as sparse? OptNet has a "lsqr" mode that significantly accelerates performance when handling sparse optimization problems. It seems the authors did not correctly utilize the different modes of the existing methods.

6. Besides, I think the authors should elaborate more on why dQP is particularly effective for large-scale problems. What mechanism specifically helps address this challenge?

If my question are well-addressed, I'll be happy to raise my score.

---

> ### Author Response · Authors · 2024-11-22
> **Official Comment by Authors - Part 1**
>
> We thank the reviewer for their evaluation. We respond to their concerns below.
>
> *Weaknesses*
>
>
> > I believe the authors do not clearly address their contribution. If the goal of this paper is to propose a generic framework, then it should be clarified how dQP incorporates other solvers. On the other hand, if dQP is intended as a specific method of differentiate opt layer, it should be compared to various state-of-the-art approaches to demonstrate its advantages.
>
> A1: We respectfully disagree with this assertion. dQP is proposed as a differentiable layer that integrates seamlessly with any QP solver in the forward pass. We describe how solvers are incorporated using the interface qpsolvers in the implementation details of section 3. We also provide more implementation details in Appendix E, including the supported solvers.
>
> Moreover, we compare dQP – namely, a variety of state-of-the-art non-differentiable QP solvers with our differentiation code attached to it – against competitor differentiable layers in section 4. These are the QP-specialized OptNet (qpth), QPLayer, and SCQPTH. We omit other generic differentiable optimization methods such as JAXopt, CVXPYLayers, and Theseus which, owing to their generality, are not best suited for QPs. This is supported in the recent works of QPLayer [1] and SCQPTH [2], where JAXopt and CVXPYLayers showed the least competitive performance. We clarify these points in the revisions of the related works, new lines 166-174.
>
> [1] Table 4, Appendix C.2.1 https://openreview.net/forum?id=YCPDFfmkFr
> [2] Sec 4.1, 4.2 https://arxiv.org/pdf/2308.08232
>
> > In Eq (10), it seems all the inequality constraints are tightened into equality constraints. How does this approach ensure the feasibility of the problem? Wouldn’t this compromise the feasibility, potentially resulting in the optimal solution of the modified optimization problem being different from the original one?
>
> A2: Only the active inequality constraints, which satisfy $c_{i}^T z^* - d_{i} = 0$ at the optimal point $z^*$, are tightened in Eq. (10). Feasibility is not compromised. Figure 3 illustrates this fact where, given the optimal point, the feasible set (a polyhedron) can be equivalently replaced with a single equality constraint, corresponding to the *active* inequality, without affecting feasibility.
>
> > However, if this is not the case and a black-box solver is used to solve the problem, then essentially a new forward process is designed for backward optimization but is not actually used, there will be two forward processes, which makes this step seem redundant.
>
> A3: The primal solution is solved for only once, using a black-box QP solver. It is only with this primal solution that the reduced system can be constructed. Eq. (6) is only used if the dual solution is not provided by the solver. No step is redundant, as detailed in Algorithm 1.
>
> > One more issue is that, in the abstract (and line 355), the authors wrote, "Our implementation, which will be made publicly available, interfaces with an existing framework that supports over 15 state-of-the-art QP solvers." Maybe I think the implementation hasn’t been fully prepared before submission, so the paper may not be fully ready for publication, as it seems there is still some unfinished work.
>
> A4: We privately shared our complete codebase with all reviewers. It will be made publicly available upon acceptance.
>
> *Questions*
>
> > What are the benefits of dQP over OptNet? Is it more efficient? If so, why? Intuitively, it seems that compared to existing methods, the number of computation steps hasn’t decreased—rather, additional steps related to tolerance testing have been introduced.
>
> and
>
> > I think the authors should elaborate more on why dQP is particularly effective for large-scale problems. What mechanism specifically helps address this challenge?
>
> A5: OptNet implements its own interior-point solver (PDIPM) and is well-suited for solving many small, dense problems via GPU batching. However, it scales poorly with problem size and it is prone to failure on ill-conditioned problems. dQP interfaces with any state-of-the-art QP solver, including those designed and optimized for large-scale, challenging problems. It fully supports sparsity in both its forward and backward passes.
>
> > Regarding the duality gap, why does OptNet show a larger duality gap in Figure 4, even though the forward method is the same for both OptNet and dQP?
>
> A6: The forward method is not the same. OptNet used their own custom primal-dual interior point method (PDIPM), whereas dQP augments various non-differentiable solvers available in qpsolvers [1], e.g. PIQP, Gurobi, ProxQP, etc. These state-of-the-art solvers typically obtain accurate solutions with smaller duality gaps.
>
> [1] https://github.com/qpsolvers/qpsolvers/#solvers

---

> > ### Author Response · Authors · 2024-11-22
> > **Official Comment by Authors - Part 2**
> >
> > > The Bi-Level Geometry Optimization in the last part of the experiment is not a common setup. I’m curious about how it can be formulated into KKT conditions. Could you provide more details on this?
> >
> >
> > A7: We do not use KKT conditions to directly tackle the entire bilevel optimization problem, which is notably nonconvex. Instead, dQP (and the forward solver) use the KKT conditions for the inner QP, whereas the outer optimization is handled by the user’s choice of iterative method for loss minimization, in this case an Adam optimizer [1]. The geometry setup described in section 4 only requires formulating the inner QP parameters from the Laplacian in the objective, constraints on vertices to the desired boundary, and constraints on the Laplacian applied at reflex vertices to satisfy the cone constraint.
> >
> > [1] Diederik P. Kingma and Jimmy Ba. Adam: A method for stochastic optimization, 2017.
> >
> > > How is Sparse QP defined in the paper, and what type of problems qualify as sparse?
> >
> > A8: For the MM dataset distributed by qpbenchmark [1], we use it as-is, which is in sparse format since many problems have low numbers of nonzeros, or fill–in ratio. The random sparse QPs, as detailed in Appendix G.1.1, are generated with $P=L^{T}L$, where $L$ is the Laplacian matrix from 3-connected random graphs. The constraints are set with a fill-in ratio 5e-4. Projection onto the probability simplex and projection onto chains, as outlined in lines 414 and lines 415 of the original manuscript, both have sparse constraint matrices and identity matrices for the quadratic terms.
> >
> > [1] https://github.com/qpsolvers/qpbenchmark
> >
> > >OptNet has a "lsqr" mode that significantly accelerates performance when handling sparse optimization problems. It seems the authors did not correctly utilize the different modes of the existing methods.
> >
> > A9: We are not aware of an “lsqr” mode for OptNet. We tested OptNet using its PDIPM method on CPU, as is also done in the previous works QPLayer and SCQPTH. OptNet’s present repository includes undocumented support for sparse problems that appears to be incomplete and failed to run in our evaluation.

---

> > > ### Comment · Reviewer_odJJ · 2024-11-28
> > >
> > > Thanks to the author for your feedback. Here are my further concerns to each question:
> > >
> > > > A1, A5, and A9:
> > >
> > > First, there is no evidence suggesting that existing models are unsuitable for QP, and I believe this is not true. Even if it were, you still need to experimentally demonstrate the effectiveness of your solver. Otherwise, given that there are already many models capable of completing this task, your work does not seem to offer significant value, that is, the novelty is not clear. Additionally, since you are working on differentiable optimization, you should compare more differentiation methods. Apart from qpth, there is also diffcp in cvxpylayer. Apologies, the "lsqr" mode should belong to it rather than OptNet. It is much more efficient than OptNet in sparse cases, but not your statement "the least competitive performance".
> > >
> > > > A2:
> > >
> > > That response does not seem convincing to me. Since the constraints have already been relaxed, how can feasibility remain unaffected? If so, it seems more theoretical proof is needed to substantiate this claim.
> > >
> > > > A3, A4:
> > >
> > > These seem good to me.
> > >
> > > > A6:
> > >
> > > What if the same forward solver is selected? It should yield the same results. Still, I firmly believe that simply replacing it with other existing forward methods is not a contribution. This point should not be highlighted.
> > >
> > > > A7:
> > >
> > > So based on your response, it seems that your model is not limited to convex problems and can also extend to non-convexity. If that is the case, this point warrants a more detailed explanation and is worth highlighting.
> > >
> > > > A8:
> > >
> > > I’m inclined to side with reviewer zgim.
> > >
> > > To summarize, I am sorry to say that I currently do not believe this paper meets the acceptance standards for ICLR. There are indeed a lot of things that need improvement. I think the authors need to supplement this work, by comparing it with more related studies, adding experimental evidence, and providing further theoretical proofs for the previously mentioned unclear aspects.

---

> ### Author Response · Authors · 2024-11-29
>
> > A1, A5, and A9: First, there is no evidence suggesting that existing models are unsuitable for QP, and I believe this is not true. Even if it were, you still need to experimentally demonstrate the effectiveness of your solver. Otherwise, given that there are already many models capable of completing this task, your work does not seem to offer significant value, that is, the novelty is not clear. Additionally, since you are working on differentiable optimization, you should compare more differentiation methods. Apart from qpth, there is also diffcp in cvxpylayer. Apologies, the "lsqr" mode should belong to it rather than OptNet. It is much more efficient than OptNet in sparse cases, but not your statement "the least competitive performance".
>
> We do not claim that “existing models are unsuitable” but rather that *generic* methods are “not *best* suited for QPs.” Our work aligns with recent studies on differentiable QP methods and we provide comprehensive evaluation against competing differentiable QP methods. CVXPYLayer has been shown to underperform compared to both QPLayer (Table 4, Appendix C.2.1 https://openreview.net/forum?id=YCPDFfmkFr) and SCQPTH (Sec 4.1, 4.2 https://arxiv.org/pdf/2308.08232) on dense problems, and we found it failed on large-scale sparse problems (i.e., crashing during setup for dim $\gtrapprox$ 2000).
>
> >A2: That response does not seem convincing to me. Since the constraints have already been relaxed, how can feasibility remain unaffected? If so, it seems more theoretical proof is needed to substantiate this claim.
>
> This idea is well-established in optimization literature and stems from complementarity. For example, see Eqs. (16.37a-d) on page 464 of Numerical Optimization by Nocedal and Wright (2006). Our proof of Theorem 1 (Appendix A) establishes the equivalence between the original QP and its reduced equality-constrained version, *thereby proving that the feasibility of the solution is unaffected*. Our proof further leverages the Basic Sensitivity Theorem (lines 238–269 in our paper) to establish a local equivalence, enabling differentiation.
>
> >A7: So based on your response, it seems that your model is not limited to convex problems and can also extend to non-convexity. If that is the case, this point warrants a more detailed explanation and is worth highlighting.
>
>
> *dQP is only for convex QPs*.
>
> Said example demonstrates how dQP can be used to approximate the solution of a non-convex bi-level optimization problem, where the inner (lower-level) problem is a convex QP. Similarly, dQP—or any other differentiable convex optimization component—can be integrated into a more complex architecture, resulting in a non-convex optimization problem (e.g., training a deep neural network that incorporates a differentiable convex QP layer).
>
>
> >A8: I’m inclined to side with reviewer zgim.
>
> We don’t understand. In our A8 we replied to your question “How is Sparse QP defined in the paper, and what type of problems qualify as sparse?” Reviewer zgim has not expressed a similar question.

---

### Official Review · Reviewer_zgim · 2024-11-01

**Soundness:** 2
**Presentation:** 3
**Contribution:** 2
**Rating:** 5
**Confidence:** 4

**Summary:**

The paper belongs to the area that focuses on embedding optimization problems into neural networks. Here, optimization algorithms are integrated directly within neural networks, allowing the optimization problem to be treated as a layer with gradients that can propagate through it. This enables end-to-end learning, where neural networks learn not only representations but also parameters of embedded optimization problems.
The paper aims to integrate any quadratic programming solver in a black-box manner without specific requirements (like the availability of a dual solution), i.e., it implements the backward pass for any optimizer, using only its primal solution. The method is based on differentiating the KKT optimality conditions and solving the reduced system, where the unnecessary inactive constraints are removed.

**Strengths:**

- The material is presented with care, and it is easy to read and follow. The problem is clearly described in great detail, and I believe the text is accessible to a broad audience.
- The idea of an independent and universal backward pass for any QP is interesting both theoretically and practically, and this paper contains a valid contribution to this topic.
- Although the idea of solving the differentiated KKT system with removed inactive constraints already appeared in prior works, this paper systematically describes and formalizes it.
- Theorem 1 allows us to compute the backward pass efficiently solely from the forward primal solution. This seems to be a new contribution.

**Weaknesses:**

- Although I personally accept the motivation that drives this work, the presented arguments should be backed by some evidence:
	- At the top of page 2, it is claimed that "many optimizers" do not provide dual solutions without mentioning any. The listed ones (Gurobi and MOSEK) provide them for QP problems.
	- There, it is also claimed that many existing methods are coupled to a tailor-made solver. Please name them.
	- At the end of related work/differentiated programming, it is claimed that the mentioned frameworks often offer limited support for solvers and may require access to a dual solution. I'm unfamiliar with all the details of these methods, and it would be beneficial if the paper could list the limitations explicitly.
- Some fragments of the theory and algorithm are missing. The overall idea is (in my opinion) clear and known. The strengths could have been in filling in the important details.
	- Computing the dual solution is said to be optional (step 4 in Algo. 1). I don't see how the derivatives (step 5) are computed if step 4 is skipped -- Eq. 7 contains both primal and dual solution.
	- Factorization (l. 320) and prefactorization (l.317, l.330) are mentioned, but never explained
- Unfortunately, any quantitative evaluation of the method is missing. All three presented experiments demonstrate and compare the performance of a selected solver. The proposed backward pass (Alg. 1) is not evaluated and I cannot confirm the proposed computational benefits of dQP.
	1. (Modularity and Performance) The "best-performing solver for each problem" with dQP backward is shown to be faster and more accurate than certain methods with their fixed solvers dedicated to certain problems. The backward pass times are not shown in separation. It it just a solver benchmark (and moreover not a fair one; OptNet was not run on GPU, which is its strength.)
	2. (Scalability) Also, OptNet was designed for lot of small dense problems and has a parallel implementation on GPU. The comparison on large sparse problems on CPU is not much valuable.
	3. (Sudoku) Loss over epoch is not informative since both methods should do the same thing (GD using analytic gradients). Training over time can tell more if dQP would prove to have more efficient backward. However (as noted) dQP suffers from infeasible solutions and OptNet is faster due to batched GPU implementation.
	4. (Bi-Level Geometry Optimization) It is just a solver comparison of forward times, not related to dQP.

Overall, the problem is well-motivated, and the approach is theoretically and practically promising. While the novelty of deriving the backward pass from the forward primal solution alone is valuable, gaps in the theoretical explanation, missing evidence to support the claims about limitations in existing methods, as well as the lack of a quantitative evaluation directly comparing the proposed backward pass with other methods, limit the paper's impact.

**Questions:**

- It is mentioned that gradients are obtained 'explicitly' from the primal solution. How?
- It is frequently promoted that something is 'explicit' ("Notably, our explicit perspective recasts the traditional implicit differentiation approach into an explicit method" l.87, "derivatives can be explicitly derived" l.272, "optimal point can be explicitly differentiated" l.301, "it provides an explicit expression for both the primal-dual optimal point" l.293). I don't see how the formulas are more explicit than those in the implicit function theorem -- It requires solving a linear system. I recommend avoiding this promotion.
- The paper claims that it is the first method of this kind. For instance, "LPGD: A General Framework for Backpropagation through Embedded Optimization Layers" unifies several methods and is applicable also to QP (they also do the Sudoku experiment)


- l.15 “This has so far prevented the use of state-of-the-art black-box numerical solvers within neural networks, as they lack a differentiable interface.” There are several solutions to this problem, mainly of LP, see the LPGD paper and the references therein.
- l. 244 “ensuring that a small perturbation of the _solution_ does not alter the active set” shouldn't it be 'parameters'?
- l. 250 "Implicit differentiation of these yields..." What is implicit here? It's just taking the derivatives of the equations and the chain rule.
- l.312"...the basic Sensitivity Theorem allows one to bypass the need for implicit differentiation techniques." If I understand correctly, the Theorem is based on the implicit function theorem. Hence, there is no bypass, in fact.
- l.322 I don't understand the use of 'linear solvers to exploit the problem's structure'.
- Fig 3 is not referred to in the text, and I don't understand its purpose.
- I think the overall idea of dQP is repeated too many times, and then the place for more valuable things is missing. It also feels like marketing in some places.
- Section "Approach" may be called standardly "Method."
- Related work might also mention "Unrolling methods."

---

> ### Author Response · Authors · 2024-11-22
> **Official Comment by Authors - Part 1**
>
> > The material is presented with care, and it is easy to read and follow. The problem is clearly described in great detail, and I believe the text is accessible to a broad audience. … The idea of an independent and universal backward pass for any QP is interesting both theoretically and practically, and this paper contains a valid contribution to this topic.
>
> We are happy the reviewer received our work positively. We indeed hope to share our work with a broader audience.
>
> *Weaknesses*
>
> > At the top of page 2, it is claimed that "many optimizers" do not provide dual solutions without mentioning any. The listed ones (Gurobi and MOSEK) provide them for QP problems.
>
> A1: Thank you, it is correct the listed ones have them and we revised our statement in the revision.
>
> > There, it is also claimed that many existing methods are coupled to a tailor-made solver. Please name them.
> and
> > At the end of related work/differentiated programming, it is claimed that the mentioned frameworks often offer limited support for solvers and may require access to a dual solution. I'm unfamiliar with all the details of these methods, and it would be beneficial if the paper could list the limitations explicitly.
>
> A2: The differentiable solvers we were referring to are QPLayer, SCQPTH, Alt-Diff, OptNet, and CVXPYLayers (diffcp). QPLayer, SCQPTH, and Alt-Diff are integrated with ProxQP, OSQP, and a custom ADMM method, respectively, both in their paper presentation and in their public implementation. OptNet’s publication and their original codebase was tightly integrated with a custom interior-point solver (PDIPM), and later minimally modified to accommodate internal calls to cvxpy with default settings for the forward pass. CVXPYLayers, through diffcp, originally supported SCS and later added support for ECOS and Clarabel. Since these methods differentiate the KKT conditions, they require the dual solution and obtain it from their forward solves directly. We clarify these points in the revisions new lines 166-174.
>
> > Some fragments of the theory and algorithm are missing. The overall idea is (in my opinion) clear and known. The strengths could have been in filling in the important details.
>
>
> A3: One of our aims was to fill in the details, as mentioned in lines 176-180. We would be happy to address specific details you believe are missing and direct us towards.
>
> > Computing the dual solution is said to be optional (step 4 in Algo. 1). I don't see how the derivatives (step 5) are computed if step 4 is skipped -- Eq. 7 contains both primal and dual solution.
>
> A4: Thank you for pointing this out, this is now clearly specified in Algorithm 1.
>
> >Factorization (l. 320) and prefactorization (l.317, l.330) are mentioned, but never explained
>
> A5: We have clarified factorization in new lines 340-344.
> >Unfortunately, any quantitative evaluation of the method is missing. All three presented experiments demonstrate and compare the performance of a selected solver. The proposed backward pass (Alg. 1) is not evaluated and I cannot confirm the proposed computational benefits of dQP.
>
> A6: We strongly disagree that quantitative evaluation is missing in our first submission. dQP’s computational advantage does not come solely from the backward pass in algorithm 1, but rather by also inheriting the efficiency of other solvers that are not differentiable without dQP.  We include additional tables of statistics for the original benchmark experiments in the revision to reinforce this point made in Figure 4 and the text.
>
> > (Modularity and Performance) The "best-performing solver for each problem" with dQP backward is shown to be faster and more accurate than certain methods with their fixed solvers dedicated to certain problems. The backward pass times are not shown in separation. It it just a solver benchmark (and moreover not a fair one; OptNet was not run on GPU, which is its strength.)
> and
> > (Scalability) Also, OptNet was designed for lot of small dense problems and has a parallel implementation on GPU. The comparison on large sparse problems on CPU is not much valuable.
>
> A7: As the reviewer has observed, we noted in our original related work (lines 147–149) that OptNet is specifically designed for small, dense problems. Our goal is not to reconfirm a well-known, published result of OptNet, but rather to show dQP’s advantage in a different, relatively untested large-scale regime. We now emphasize this in the revised experiments section (new lines 393-400), pointing out that dQP in its present CPU-only form thus complements, not supersedes, OptNet. The tables in the revised version include the backward pass time in isolation for clear evaluation.

---

> > ### Comment · Reviewer_zgim · 2024-11-27
> >
> > Thanks again for the responses.
> >
> > I want to take back the weakness that 'any quantitative evaluation of the method is missing'. My judgment was based on the fact that dQP is not the only black-box method, and I was focused mainly on the contribution in the backward pass computation, which was not, from my perspective, evaluated properly. (And I also misinterpreted the last experiment.) I accept your perspective of speeding up the whole pipeline by a custom selection of the best-performing solver, which you focused on in the evaluation.
> > I raised the rating.
> >
> > Thanks also for adding the results in Table 1 with backward times reported.

---

> ### Author Response · Authors · 2024-11-22
> **Official Comment by Authors - Part 2**
>
> > (Sudoku) Loss over epoch is not informative since both methods should do the same thing (GD using analytic gradients). Training over time can tell more if dQP would prove to have more efficient backward. However (as noted) dQP suffers from infeasible solutions and OptNet is faster due to batched GPU implementation.
>
> A8: We have opted to include Sudoku as a classic proof-of-concept learning experiment. We do not time the experiment because OptNet is suited for GPU batching many small, dense problems (like Sudoku), solidly outside the regime where dQP aims to make significant speed gains. Moreover, we note previous works, e.g. QPLayer and Alt-Diff, similarly perform all of their experiments on CPU.
>
> > (Bi-Level Geometry Optimization) It is just a solver comparison of forward times, not related to dQP.
>
> A9: While only forward times are strictly compared in this experiment, the execution of the bi-level optimization hinges on the derivatives, in this case of the dual variables at the optimal point, to perform gradient descent in the outer problem. OptNet and SCQPTH do not expose their duals in their codebase, nor differentiate them, thus we could only evaluate their forward pass performance, as an indication of their potential overall performance.
> *Questions*
>
> > l. 250 "Implicit differentiation of these yields..." What is implicit here? It's just taking the derivatives of the equations and the chain rule.
>
> and
>
> > l.312"...the basic Sensitivity Theorem allows one to bypass the need for implicit differentiation techniques." If I understand correctly, the Theorem is based on the implicit function theorem. Hence, there is no bypass, in fact.
>
> and
>
> > It is mentioned that gradients are obtained 'explicitly' from the primal solution. How?
>
> and
>
> > It is frequently promoted that something is 'explicit' ("Notably, our explicit perspective recasts the traditional implicit differentiation approach into an explicit method" l.87, "derivatives can be explicitly derived" l.272, "optimal point can be explicitly differentiated" l.301, "it provides an explicit expression for both the primal-dual optimal point" l.293). I don't see how the formulas are more explicit than those in the implicit function theorem -- It requires solving a linear system. I recommend avoiding this promotion.
>
> A10: Our use of “implicit differentiation” here is referring to the process of differentiating an implicit function, which, to our knowledge, is classic terminology and is used e.g. in [1]. Likewise, we refer to ordinary differentiation as explicit differentiation to clearly distinguish them. In detail: implicit differentiation is applied to the full nonlinear KKT conditions, since they define the solution implicitly. In contrast, explicit differentiation is applied to the closed-form expression for the solution to the reduced linear KKT conditions, since their reduced solution is defined explicitly.
> While we agree that the implicit function theorem is used to establish the basic sensitivity theorem, our claim is that the implicit differentiation technique, as aforementioned, is no longer needed. Namely, Theorem 1 shows that the QP in Eq. (1) and its reduced form in Eq. (5), which defines the reduced solution explicitly, are equivalent in a neighborhood of the parameter $\theta$. Hence, the expressions for derivatives obtained by these two approaches applied to their respective problem must be the same, and we do not suggest otherwise.
> We believe this observation and distinction make it clearer that the (reduced) derivatives are “just” that of a symmetric linear system, which also shares its matrix with the system for the derivative.
> We add these clarifications to the revision.
> We hope this justifies our use of “explicit” in all of the instances raised.
>
> [1] Page 9 Steven G. Krantz and Harold R. Parks. The Implicit Function Theorem. Birkhauser Boston, MA, 2012.
>
> > The paper claims that it is the first method of this kind. For instance, "LPGD: A General Framework for Backpropagation through Embedded Optimization Layers" unifies several methods and is applicable also to QP (they also do the Sudoku experiment)
>
> and
>
> > “This has so far prevented the use of state-of-the-art black-box numerical solvers within neural networks, as they lack a differentiable interface.” There are several solutions to this problem, mainly of LP, see the LPGD paper and the references therein.
>
> A11: Thank you for pointing us to this recent interesting paper, we have added it to our list of general methods in the revised related works (new line 165). We did not intend to directly claim to be the first general differential framework that supports many solvers and mention the availability of such work (e.g., line 164-line 167). We soften the language in our revised abstract and introduction to make more precise statements. (e.g. “prevented” to “limited” line 15 as noted).

---

> > ### Author Response · Authors · 2024-11-22
> > **Official Comment by Authors - Part 3**
> >
> > > l. 244 “ensuring that a small perturbation of the solution does not alter the active set” shouldn't it be 'parameters'?
> >
> > A12: Thank you, we have fixed this.
> >
> > > l.322 I don't understand the use of 'linear solvers to exploit the problem's structure'.
> >
> > A13: As briefly alluded to in lines 361–364 of the original manuscript, we considered two specific structures: (1) The reduced system for the derivatives takes the form of a classic block symmetric indefinite KKT matrix, which frequently arises in optimization. Specialized methods, such as QDLDL (incorporated in our software package), are designed to solve such systems efficiently. (2) Problems that result in structured sparse systems, such as those involving Laplacian matrices of graphs, can leverage tailored solvers. We have clarified this point in the revised text at line 322.
> >
> > > Fig 3 is not referred to in the text, and I don't understand its purpose.
> >
> > A14: The purpose of Fig 3 is to illustrate differentiation and the local equivalence between the original and reduced problems. It demonstrates that, at the optimal point, the feasible set (a polyhedron) can be equivalently replaced with a single equality constraint corresponding to the *active* inequality without affecting feasibility or the local behavior (sensitivity) of the optimal point. We add a reference in the main text of the revision.
> >
> > > Related work might also mention "Unrolling methods."
> >
> > A15: Our original submission includes a comment on unrolling methods lines 129-l.131. Please let us know if there is additional work you want us to include.

---

### Official Review · Reviewer_Jnqb · 2024-11-02

**Soundness:** 3
**Presentation:** 3
**Contribution:** 2
**Rating:** 6
**Confidence:** 4

**Summary:**

This paper proposes dQP, a framework for differentiation of blackbox quadratic programming solvers. The method is based on the observation that an inequality-constrained QP can be re-phrased to a locally equivalent equality-constraint QP given the active set at the solution. This eliminates the inactive inequality constraints from the differential KKT matrix, which is substantially reduced in dimension and becomes symmetric, allowing for faster derivative computation. The required factorization can also be used to compute dual variables from the primal variables, enabling support for solvers that only produce primal variables as outputs.
Experimentally, the authors test their method on a diverse benchmark of QP problems. Especially when the QPs are structured, dQP inherits the benefits of the superiority of the enabled blackbox solvers compared to the simpler dense solvers in existing differentiable QP layers, which allows dQP to be applied to much larger sparse QPs.

**Strengths:**

- dQP as a programming package seems very useful, the ability to choose between any state-of-the-art solver is a strong selling point.
- Re-using the matrix factorization that from the derivative computation for calculating the optimal dual variables from the primal variables is useful insight.
- The bi-level geometry optimization problem is an interesting new application of differentiable optimization methods.

**Weaknesses:**

- The theoretical contribution of the paper is minor. The only insight is that the optimal dual variables can be obtained by re-using the KKT matrix factorization.
- The experiments do not sufficiently disentangle forward and backward computation time. Whenever the forward+backward time is reported, the authors should also report the the forward and backward time individually. It is clear that the solvers used in dQP can solve larger problems faster on the forward pass, but it is also important to see how the required compute for the derivative computation compare to exisitng differentiable QP layers.
- Currently, the "modularity and performance" and "scalability" experiments feel like simply benchmarking the different solvers, the gradient computation time is reported but no actual learning takes place. The Sudoku experiment does not really show any advantage over the previous OptNet method (also, here the training times are missing to jusge how big the disadvantage of not allowing GPU parallelization in dQP is). The paper would be much stronger if these results would be moved to the appendix, and more results of the kind of "Bi-Level Geometry Optimization" would be included instead.
- dQP does not allow GPU parallelization, as opposed to previous methods.

**Questions:**

- I am confused regarding the authors interpretation of explicit vs implicit, which is highlighted many times throughout the manuscript. If I understand correctly, the authors use the term explicit to refer to the fact that by using a factorization of the KKT matrix inverse (e.g. eq (6) and (7)) no optimization procedure has to be used to obtain the result on the left hand side. But why is this in any way fundamentally different to the (according to the authors implicit) differentiation in eq. (3) by multiplying with the inverse of the KKT matrix and solving using a factorization? After all, eq. (7) is exactly the result obtained from the *implicit* function theorem applied to an equality-constrained QP?
- The factorization used to solve eq (6) and (7) depends on the active set $J$, which means we cannot re-use it if we encounter a different active set by changing only the vectors $(q,b,d)$ while keeping $(P, A, C)$ constant. This is in contrast to computing a factorization of the whole KKT matrix as in OptNet, which can be re-used for any value of $(q,b,d)$. Have the authors thought about this potential drawback? I assume it could be especially relevant in some applications where $(P, A, C)$ is considered constant throughtout the training procedure and only $(q,b,d)$ is learned. Is it e.g. possible to re-use the factorization of the KKT matrix with $(P, A, C_J)$ to more efficiently compute the factorization of a KKT matrix for a different active set $(P, A, C_{J'})$?
- The authors write: "Furthermore, our approach not only simplifies computation but also allows for the use of fast, specialized linear solvers that exploit the problem’s structure" (line 322) Could the authors elaborate on this? Is this already used in the software package?

---

> ### Author Response · Authors · 2024-11-22
> **Official Comment by Authors - Part 1**
>
> > dQP as a programming package seems very useful, the ability to choose between any state-of-the-art solver is a strong selling point.
>
> We thank the reviewer for the positive feedback. Our hope is that dQP’s modularity makes it easier for the community to incorporate differentiable QPs in their work.
>
> *Weaknesses*
>
> > The experiments do not sufficiently disentangle forward and backward computation time. Whenever the forward+backward time is reported, the authors should also report the the forward and backward time individually. It is clear that the solvers used in dQP can solve larger problems faster on the forward pass, but it is also important to see how the required compute for the derivative computation compare to existing differentiable QP layers.
>
> A1: We address this point in our revision by including tables of forward and backward statistics in the main text and appendix.  To summarize, for smaller dense problems (MPC dataset), as reported in Figure 4 and new Table 1, differentiation tightly integrated with a solver (e.g., in QPTH, SCQPTH, QPLayer) will typically outperform our decoupled differentiation, as anticipated. However, we show that even if dQP’s backward pass is less efficient, the gains dQP achieves by using state-of-the-art solvers for the forward pass can be sufficient to make dQP the overall fastest, in addition to being the most accurate. This advantage is especially prominent in the large-scale datasets, as illustrated in updated Figure 1 (previously Figure 10).
>
> > dQP does not allow GPU parallelization, as opposed to previous methods.
>
> A2: dQP is indeed limited to CPU, similar to other previous methods like QPLayer and Alt-Diff. For dQP, this stems from the absence of state-of-the-art QP solvers effectively supporting GPU. Notably, dQP performs particularly well on large, sparse problems that may not be well-suited to GPU batching. Thus, dQP and competing GPU batchable methods such as OptNet best operate in different problem regimes, complementing one another. We emphasize this in the revision in the new lines 393-400.
>
> *Questions*
>
> > … If I understand correctly, the authors use the term explicit to refer to the fact that by using a factorization of the KKT matrix inverse (e.g. eq (6) and (7)) no optimization procedure has to be used to obtain the result on the left hand side. But why is this in any way fundamentally different to the (according to the authors implicit) differentiation in eq. (3) by multiplying with the inverse of the KKT matrix and solving using a factorization? After all, eq. (7) is exactly the result obtained from the implicit function theorem applied to an equality-constrained QP?
>
> A3: Implicit and explicit differentiation are terminological distinctions that refer to the function being differentiated and the approach used to derive its derivative. In detail: implicit differentiation [1] is applied to the full nonlinear KKT conditions, since they define the solution implicitly. In contrast, explicit differentiation is applied to the closed-form expression for the solution to the reduced linear KKT conditions, since their reduced solution is defined explicitly.
> Using the sensitivity theorem, Theorem 1 shows that the QP in Eq. (1) and its reduced form in Eq. (5) are equivalent in a neighborhood of the parameter $\theta$. Hence, the expressions for derivatives obtained by these two methods must be the same, and we do not suggest otherwise.
> We believe this observation and distinction make it clearer that the (reduced) derivatives are “just” that of a symmetric linear system, which also shares its matrix with the system for the derivative.
> We added these clarifications to the revision, new lines 319-338.
>
> [1] Page 9 Steven G. Krantz and Harold R. Parks. The Implicit Function Theorem. Birkhauser Boston, MA, ¨ 2012.

---

> > ### Author Response · Authors · 2024-11-22
> > **Official Comment by Authors - Part 2**
> >
> > > The factorization used to solve eq (6) and (7) depends on the active set
> > $J$, which means we cannot re-use it if we encounter a different active set by changing only the vectors $(q,b,d)$ while keeping $(P,A,C)$ constant. This is in contrast to computing a factorization of the whole KKT matrix as in OptNet, which can be re-used for any value of  $(q,b,d)$. Have the authors thought about this potential drawback? I assume it could be especially relevant in some applications where $(P,A,C)$ is considered constant throughtout the training procedure and only $(q,b,d)$ is learned. Is it e.g. possible to re-use the factorization of the KKT matrix with $(P,A,C_J)$ to more efficiently compute the factorization of a KKT matrix for a different active set  $(P,A,C_{J’})$?
> >
> > A4: We have considered the possibility of reusing information between consecutive QP solves. For instance, classic active set methods use factorization updates as few constraint rows change [1], an idea that may be adapted if outer iterations (e.g., in learning or bilevel optimization), did not change the active set significantly. Since this is not generally guaranteed, we were not motivated to pursue this for our present work, though it is a promising idea for specialized problems.
> >
> > OptNet incorporates an internal interior-point QP solver which uses partial factorization in its iterations for estimating the optimal point. It further re-uses this factorization to compute the derivatives, thus reducing its backward pass time. While OptNet could in principle perform a partial factorization if some parameters are fixed between consecutive QP solves, they do not to the best of our knowledge.
> >
> > [1] Page 478 in Jorge Nocedal and Stephen J Wright. Numerical optimization, 2006
> >
> > > The authors write: "Furthermore, our approach not only simplifies computation but also allows for the use of fast, specialized linear solvers that exploit the problem’s structure" (line 322) Could the authors elaborate on this? Is this already used in the software package?
> >
> > A5: We thank the reviewer for pointing this out. As briefly alluded to in lines 361–364 of the original manuscript, we considered two specific structures: (1) The reduced system for the derivatives takes the form of a classic block symmetric indefinite KKT matrix, which frequently arises in optimization. Specialized methods, such as QDLDL (incorporated in our software package), are designed to solve such systems efficiently. (2) Problems that result in structured sparse systems, such as those involving Laplacian matrices of graphs, can leverage tailored solvers. We have clarified this point in the new lines 351-352.

---

> > > ### Author Response · Authors · 2024-12-02
> > >
> > > Dear Reviewer Jnqb,
> > >
> > > Please let us know if we have addressed your questions or if there are any additional concerns we can clarify before the discussion period ends.

---

> ### Comment · Reviewer_Jnqb · 2024-12-03
>
> Thank you for your responses, I have decided to raise my score.

---

### Official Review · Reviewer_3S65 · 2024-11-02

**Soundness:** 2
**Presentation:** 3
**Contribution:** 2
**Rating:** 5
**Confidence:** 4

**Summary:**

This paper addresses a common issue in Quadratic Programming (QP) by proposing a decoupled algorithm for computing gradients. The core principle is to utilize the KKT (Karush-Kuhn-Tucker) conditions to recover the dual variables $\lambda$, $\mu$, given only the primal variables. Additionally, the paper employs the active set method, allowing for the reconstruction of an equivalent new problem based on the provided primal solution $z^*$, which includes only the original equality constraints and the active inequality constraints. Consequently, this work presents a generalized scheme for backpropagation in computational optimization layers, expanding the options for solvers.

**Strengths:**

1. The paper achieves a commendable integration and decoupling of algorithms. Previous works in differentiable quadratic optimization often entangled forward and backward algorithms, or required backward algorithms to utilize dual variables, which were only provided by a few solvers (like gurobi and cplex). By leveraging the KKT conditions to recover dual variables and offering a corresponding backpropagation method, this paper resolves the coupling issue present in earlier approaches.

2. By addressing the coupling of forward and backward processes, the paper enables the selection of optimal solvers to enhance forward performance in the optimization layer. The extensive experiments presented demonstrate improved success rates and times in the forward process when using existing differentiable optimizers such as QPTH and SCQPTH. Thus, the findings of this paper can be applied to larger problem sizes.

**Weaknesses:**

1. While this work builds upon established methods effectively, there may be room for further originality in certain aspects. Though it successfully resolves the coupling issue in related works, it is limited to the specific case of QP. The recovery of KKT conditions in QP simplifies to solving a linear equation; however, in more general cases, altering the function (f) could lead to a non-linear problem, necessitating solvers that handle such complexities, which may incur additional computational costs.

2. There are concerns regarding the fairness of the experimental comparisons. Although the paper demonstrates faster solver speeds due to the ability to select optimal solvers, it should be noted that these optimal solvers (e.g., Gurobi) are implemented in C/C++, while QPTH and SCQPTH are implemented in Python. Given the inherent speed differences between these languages, the observed performance advantage may stem from the implementation language rather than the algorithms themselves. It would be beneficial for the authors to present additional experimental data showcasing the time taken by QPTH, SCQPTH, and dQP during both the forward and backward processes. A separate demonstration using QPTH/SCQPTH for the forward process combined with dQP for the backward process would further ensure a fair comparison.

3. The paper mentions a lack of GPU support. From the perspective of the backward process, the proposed backpropagation algorithm appears straightforward to apply to batches of data, similar to the capabilities of OptNet in this context. However, the absence of GPU support seems to arise from the external solvers utilized, which do not support GPU implementation. Given the significance of batch processing on GPU for deep learning training, it may be advantageous to develop Python versions of some solvers to enable batch mode support.

**Questions:**

See weakness.

---

> ### Author Response · Authors · 2024-11-22
>
> > The paper achieves a commendable integration and decoupling of algorithms. … the paper enables the selection of optimal solvers to enhance forward performance in the optimization layer.
>
> We thank the reviewer for their positive comments on our work.
>
> *Weaknesses/Questions*
>
> > The recovery of KKT conditions in QP simplifies to solving a linear equation; however, in more general cases, altering the function (f) could lead to a non-linear problem, necessitating solvers that handle such complexities, which may incur additional computational costs.
>
> A1: Our work focuses on QPs. By decoupling optimization and differentiation for QPs, which are prevalent in convex optimization, we have taken a significant step towards decoupling in general optimization problems. We agree that more general objectives and constraints introduce new challenges and require non-trivial adaptations to solve. We will address general optimization problems in follow-up work, building on the foundation of our work on dQP.
>
> > Although the paper demonstrates faster solver speeds due to the ability to select optimal solvers, it should be noted that these optimal solvers (e.g., Gurobi) are implemented in C/C++, while QPTH and SCQPTH are implemented in Python. Given the inherent speed differences between these languages, the observed performance advantage may stem from the implementation language rather than the algorithms themselves … A separate demonstration using QPTH/SCQPTH for the forward process combined with dQP for the backward process would further ensure a fair comparison.
>
> A2: We agree that to a large extent the advantage of highly optimized solvers comes from implementation details and language. A key strength of dQP is that it is possible to leverage these advantages and differentiate the solution, unlike tightly integrated alternatives that rely on their custom forward.
>
> Differentiation in a tightly integrated solver (e.g., in QPTH, SCQPTH, QPLayer or others) can outperform our decoupled differentiation. However, our experiments demonstrate that it often pays off to use dQP alongside a state-of-the-art non-differentiable QP solver in terms of total time and accuracy. To ensure a fair comparison of the backward alone, independent of forward performance, we include separated forward/backward timings in the revision.
>
> > …the absence of GPU support seems to arise from the external solvers utilized, which do not support GPU implementation. Given the significance of batch processing on GPU for deep learning training, it may be advantageous to develop Python versions of some solvers to enable batch mode support.
>
> A3: We agree with this desire. While adapting dQP to parallel differentiation on GPU is relatively simple, the limitation indeed stems from the absence of highly-optimized QP solvers effectively supporting GPU. Notably, dQP performs well on large, sparse problems which GPU batching may not be well-suited to. Thus, dQP and competing methods best operate in different problem regimes, complementing one another. We emphasize this in the revision in the new lines 393-400.

---

> > ### Comment · Reviewer_3S65 · 2024-11-28
> > **Official Comment by Reviewer 3S65**
> >
> > Thanks for your responses. The reviewer would suggest the authors use red or blue color to mark the specific revision. Otherwise, it is hard to track the revision. In addition, it seems like the authors added a fair comparison. Please add some analysis after that, and please also indicate where it is.

---

> > > ### Author Response · Authors · 2024-11-28
> > >
> > > We have updated the revision to highlight all changes in blue. Apologies for not doing so earlier—we had assumed OpenReview's PDF compare feature would automatically handle this. We are not entirely sure which analysis you are referring to, and it seems that we won't be able to make further changes to the manuscript. However, the additional statistics provided in Table 1 (and Tables 2–5 in the Appendix) further support the experimental results and conclusions presented and discussed in the original submission.

---

> > > > ### Author Response · Authors · 2024-12-02
> > > >
> > > > Dear Reviewer 3S65,
> > > >
> > > > Please let us know if we have addressed your questions or if there are any additional concerns we can clarify before the discussion period ends.

---

> > > > > ### Comment · Reviewer_3S65 · 2024-12-03
> > > > > **Official Comment by Reviewer 3S6**
> > > > >
> > > > > Thanks for the response and efforts in revising the paper. Together with other reviewers' comments, the reviewer intends to maintain the score.

---

### Meta-Review · Area_Chair_7YaV · 2024-12-22

**Metareview:**

Summary: The paper presents dQP, a framework for differentiating blackbox quadratic programming solvers, which allows for the integration of various solvers into neural networks without specific requirements like providing dual solutions. The method leverages KKT conditions to recover dual variables and presents a generalized scheme for backpropagation in computational optimization layers. The experiments indicate that dQP can handle larger sparse problems more efficiently than existing differentiable QP layers.

Strengths:

dQP offers a novel approach to integrating optimization problems into neural networks by differentiating blackbox solvers, which expands the range of available solvers for such tasks.

The framework's ability to work with only primal solutions is a significant advantage, as it broadens the applicability of differentiable optimization layers to solvers that do not provide dual solutions.

Drawbacks:

The paper lacks a clear comparison to state-of-the-art methods, particularly cvxpy, which is known for its efficiency in differentiating sparse problems.

The experiments do not sufficiently demonstrate the computational efficiency of dQP, with some results showing minimal differences or even underperformance compared to existing solvers.

There is a concern about the practical performance of dQP, especially regarding the potential impact of the tolerance parameter on feasibility and the lack of GPU support, which is crucial for deep learning applications.

Based on the above points , I must reject this work due to the insufficient demonstration of its advantages over existing methods, the lack of clarity on its contributions, and the concerns regarding its practical performance and applicability.

**Additional Comments On Reviewer Discussion:**

The concerns raised by the reviewers are not well addressed.

---

### Decision · Program_Chairs · 2025-01-22

Reject